**EMBO** *reports*

# Evolution of intrinsically disordered regions in vertebrate galectins for phase separation

Yu-Hao Lin [ID] [1,2,5], Yu-Chen Chen[1,5], Yung-Chen Sun[1,2] & Jie-rong Huang [ID] [1,3,4 ✉]

## Abstract

**Intrinsically disordered regions (IDRs) are widespread in proteins, yet their evolutionary paths remain poorly understood. Using galectin, a universal carbohydrate-binding protein, we investigated how IDRs evolved and acquired their biological roles in vertebrates. Through extensive proteome-wide sequence analyses, we found that vertebrate galectin IDRs share overall amino acid compositions but differ significantly in their aromatic residue types. Using nuclear magnetic resonance (NMR) spectroscopy and lipopolysaccharide micelle assays, we demonstrated that despite these differences, IDRs from various vertebrate galectins independently converged toward a similar function: mediating agglutination via phase separation. Our data suggest that the specific types of aromatic residues within these IDRs were established early in evolution and underwent independent expansions among different vertebrate lineages. Additionally, we identified a conserved short N-terminal motif critical for promoting galectin self-association, which likely served as an incipient sequence for subsequent IDR evolution. Contrary to previous peptide studies emphasizing aromatic residue specificity, our findings highlight the evolutionary preference for increasing motif repetition over residue-type optimization to achieve functional fitness.**

**Keywords** Liquid-Liquid Phase Separation; Intrinsically Disordered Proteins; Protein Evolution; NMR; Galectin
**Subject Categories** Evolution & Ecology; Structural Biology

## Introduction

Proteins' functions are determined by their unique shapes, encoded in their amino acid sequences. The sequences have been shaped by natural selection to adapt new functions or optimize existing ones, varying through evolution (Jayaraman et al, 2022). Regardless of the sequence diversity, conserved patterns hint at the existence of common ancestors (Capra and Singh, 2007), and levels of similarity between protein sequences reveal phylogenetic relationships and evolutionary routes (Zuckerkandl and Pauling, 1965). Protein evolutionary paths also provide information to predict tertiary structures from their sequences as exploited in classical homology modeling (Waterhouse et al, 2018) and in recent machine learning approaches (Jumper et al, 2021). However, intrinsically disordered proteins (IDPs) or proteins with intrinsically disordered regions (IDRs) exhibit limited homolog sequence similarity and are structurally unpredictable (Akdel et al, 2022), making their emergence evolutionarily intriguing.

At least half of the eukaryotic proteins have IDRs (Dunker et al, 2000). Their functions include increasing binding surface areas, facilitating post-translational modifications, serving as versatile hubs for interactions with multiple partners, and acting as linkers in multiple folded domains (Holehouse and Kragelund, 2024). Another key characteristic of proteins with IDRs is their ability to phase separate (PS), which is the main mechanism of membrane-less organelle (biomolecular condensate) formation for the spatiotemporal control in many biological functions (Banani et al, 2017; Shin and Brangwynne, 2017). The "spacer" and multivalent "sticker" motifs borne by the IDRs in these proteins, respectively, provide the flexibility and adhesive properties required to drive PS (Harmon et al, 2017; Holehouse and Pappu, 2018). However, without structural constraints, the evolutionary paths to these various functions remain unclear and challenging to study.

IDRs may have coevolved with folded partners by enhancing affinities at binding interfaces (Jemth et al, 2018; Karlsson et al, 2022; Mihalic et al, 2024). The length of linker IDRs connecting structural domains is also adaptable to optimize the binding affinity (Gonzalez-Foutel et al, 2022). In our previous work on a paralog protein family, we demonstrated that different physicochemical properties can converge to the same function (Chiu et al, 2022). Using ortholog databases, we also demonstrated that aromatic residues are more likely to appear in the IDRs of RNA-binding proteins to promote PS (e.g., in stress granules) (Ho and Huang 2022). Furthermore, selective pressures seem to preserve PS propensity in certain IDR-rich proteins, such as the FET protein family (Dasmeh et al, 2021; Dasmeh and Wagner, 2021) and DEAD-box ATPases (Hondele et al, 2019). Protein ortholog databases also provide valuable training datasets for machine learning to detect IDR features (Lu et al, 2022; Ho et al, 2023). However, these databases are mainly based on sequence or

[1]Institute of Biochemistry and Molecular Biology, National Yang Ming Chiao Tung University, No. 155 Section 2 Li-nong Street, Taipei, Taiwan. [2]Taiwan International Graduate Program in Molecular Medicine, National Yang Ming Chiao Tung University and Academia Sinica, Taipei, Taiwan. [3]Department of Life Sciences and Institute of Genome Sciences, National Yang Ming Chiao Tung University, No. 155 Section 2 Li-nong Street, Taipei, Taiwan. [4]Institute of Biomedical Informatics, National Yang Ming Chiao Tung University, No. 155 Section 2 Li-nong Street, Taipei, Taiwan. [5]These authors contributed equally: Yu-Hao Lin, Yu-Chen Chen. ✉E-mail: jierongh@nycu.edu.tw

functional similarities in folded domains, and the evolution routes of IDRs remain relatively less explored.

Here, we address this issue by examining galectin-3, a β-galactoside-binding lectin featuring a folded carbohydrate-recognition domain (CRD) and an intrinsically disordered N-terminal domain (NTD). In our previous biophysical and structural studies, we showed that NTDs self-associate and make fuzzy contacts with the CRD's non-carbohydrate-binding site (hereafter, the NTD-binding face) in both intra- and intermolecular modes (Fig. EV1) (Lin et al, 2017). These interactions drive galectin-3 phase separation via π–π interactions and explain that the protein's agglutination arises not from dimeric or tandem CRD repeats but from multivalent NTD-mediated association (Fig. EV1) (Chiu et al, 2020). We further assessed cation-π interactions contributed by conserved, positively charged residues at the NTD-binding face (Fig. EV1) (Sun et al, 2024). This detailed mechanistic framework makes galectin-3 a suitable model for our analysis. In this work, using proteome-wide sequence analysis, biophysical assays, and nuclear magnetic resonance (NMR) spectroscopy, we found that IDR-tethered galectins are widespread across diverse taxa, including most animals. In vertebrates, although galectin IDR sequences are similar in composition, they differ in the types of aromatic residues and their flanking motifs. Our results indicate that the agglutination capacity of these aromatic residues depends primarily on the number rather than the specific type of aromatic sequence motifs in the IDRs. This finding underscores the adaptability of these proteins and provides insights into the intriguing evolutionary trajectories of IDRs.

# Results

## Proteome-wide analysis of aromatic-containing motifs in galectins with intrinsically disordered regions

In the orthologous matrix (OMA) database (Altenhoff et al, 2021), ~100 proteins are annotated as orthologous to human galectin-3 (hGal3, OMA group number: 1215955). As expected, their amino acid compositions and sequence-predicted levels of the structural disorder are similar (Dataset EV1). However, to investigate how widespread IDR-bearing galectins are in the entire proteome, we searched for homologs using only the human CRD sequence. We used the BLAST algorithm (Zaru et al, 2023) to query the TrEMBL database (UniProt, 2023) (~200 million sequences) on a local computer with an *E*-value criterion of 0.1. This search returned ~17,000 proteins with at least 21% sequence identity to the CRD of hGal3. Among these, 1321 proteins contain at least 75 consecutive residues predicted to be disordered (disorder score >0.5) by IUPRED3 (Erdos et al, 2021) (Fig. 1A). Dataset EV2 shows the predicted disordered regions, amino acid composition, and the phylogenetic relations of these 1321 protein sequences. Representative examples from each phylum are depicted in Fig. 1B. Notably, the amino acid compositions of these IDRs differ among phyla and exhibit little sequence conservation, suggesting that IDRs tethered to galectin CRDs likely arose independently in different lineages (also see Discussion). For instance, IDRs in nematodes (e.g., *C. elegans*) and arthropods (e.g., *Drosophila*) contain higher proportions of negatively charged amino acids (aspartate and glutamate),

whereas chordate IDRs (including those in vertebrates) are enriched in proline, glycine, and alanine (Fig. 1B).

This similarity of amino acid patterns among vertebrates is in keeping with previous observations in model organisms (Chiu et al, 2020), whereby we found that the functional role of the aromatic residues in hGal3's IDR is to drive agglutination (Chiu et al, 2020). Although the IDRs have similar compositions, the types of aromatic residues present differ between vertebrates (Fig. 1C shows the distributions of %W, %F, and %Y in IDRs across vertebrates, alongside representative pie charts from Dataset EV3). For instance, aromatic residue compositions are more variable in fish galectins' IDRs than in those of amphibians, while tyrosine predominates in mammals. Of the 1321 sequences with predicted IDRs longer than 75 amino acids, 351 of them share an IDR amino acid composition analogous to hGal3, namely >8% aromatic residues (phenylalanine, F; tyrosine, Y; tryptophan W) and <3% negatively charged residues (D, E; Fig. 1A), a subset manageable for motif searches using the MEME suite web server (Bailey et al, 2015). Detailed information on their predicted disorder, amino acid compositions of these IDRs, and phylogenetic trees is available in Dataset EV3. The MEME results for all sequences are shown in Fig. 1D in a compact format for an overview. An accompanying HTML file in Dataset EV4 provides interactive features for more detailed motif information. For visual clarity, ~20 representative sequences spanning different taxonomic groups were selected (Fig. 1E; Appendix Fig. S1A). Comparing sequence motifs revealed many similarities in the CRD between species, as expected (Fig. 1D,E; same color blocks between different species, identified by BLAST). Notably, many motifs also occur multiple times within the same IDR (Fig. 1D,E; same color blocks within each sequence), several of which are shown in Fig. 1F. Furthermore, all the analyzed sequences have similar motifs at the N-terminus (Fig. 1D,E), and three of which are mapped onto residues 5–20 of hGal3 in Fig. 1G. From the motif patterns (Fig. 1E), we sought to understand whether the repeats in galectin IDRs share a common ancestral origin or if they emerged through multiple duplications of a shorter motif (Fig. 1H). The IDR-tethered galectin in zebrafish (zfGal) is particularly interesting because of the predominance of tryptophan-based motifs, in contrast to the tyrosine-based motifs in hGal3 (Fig. 1I; Appendix Fig. S1B). The respective predominance of tryptophan and tyrosine does not reflect a difference in the distribution of amino acids in the corresponding proteomes (Appendix Fig. S1C). Consequently, we selected this zebrafish galectin as a model system to experimentally investigate the effects of particular aromatic amino acids and explore possible evolutionary routes.

## Zebrafish galectin agglutination

Zebrafish galectin has similar structural disorder and prion-likeness scores to those of hGal3 (Fig. 2A), and the AlphaFold-predicted structure of the CRD is similar to the experimentally determined structure of hGal3's CRD (PDB code: 2NMO; Fig. 2B). Furthermore, circular dichroism (CD) spectra analysis of zfGal (purified from bacteria with synthetic cDNA) showed similar secondary structure populations as in hGal3 (Fig. 2C,D).

One of hGal3's functions is to agglutinate glycosylated molecules through PS via its IDR (Chiu et al, 2020). Comparable predicted levels of π-π interactions and PS propensities in zfGal and hGal3 suggest their similar condensation ability (Fig. 2E). Zebrafish galectin agglutination was experimentally investigated using lipopolysaccharide (LPS) micelle

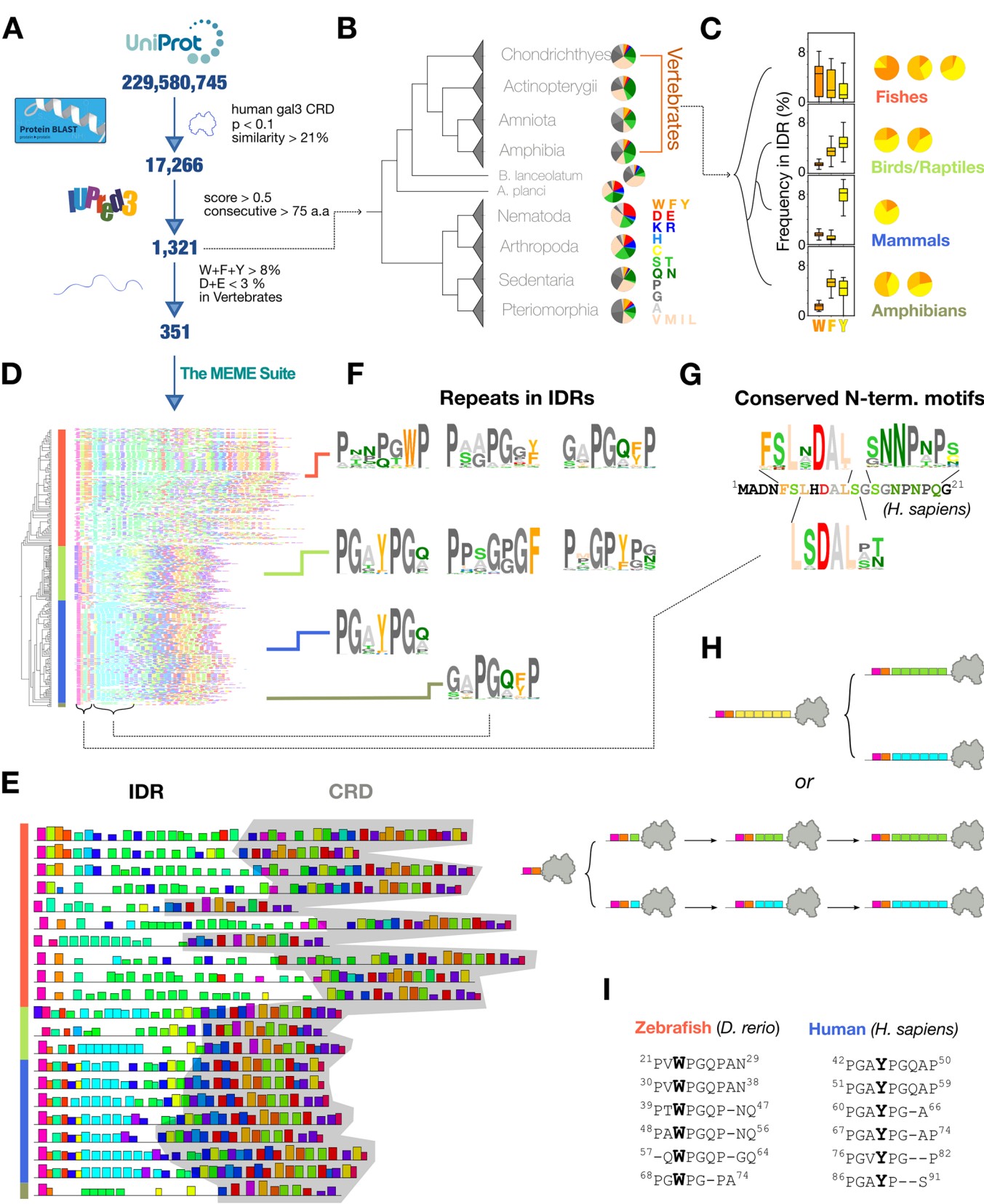

Figure 1.   Sequence analysis of galectins with long (>75 amino acids) intrinsically disordered regions (IDRs).

(A) Flowchart of the analysis with algorithms and selection criteria. (B) Taxonomical grouping of galectins with IDRs longer than 75 amino acids. Representative amino acid distributions are shown as pie charts. (C) Boxplots show the distributions of W, F, and Y content (percent of IDR) in galectin homologs across major vertebrate clades (median and interquartile range shown; whiskers denote 1.5 × IQR). Representative pie charts further illustrate the aromatic composition per group (W: tryptophan, F: phenylalanine, Y: tyrosine; all data in Dataset EV3). (D) Sequence motifs identified by the MEME suite (Bailey et al, 2015). The color blocks in each sequence represent frequently appearing motifs (see Dataset EV4 for additional information). (E) Representative MEME results selected from panel D. The carbohydrate-recognition domain (CRD) is shaded gray. (F) Representative repeated motifs in IDRs. (G) N-terminal conserved sequences aligned with human galectin-3. (H) Schematic illustration of possible evolutionary models. (I) Aligned repeated sequences in zebrafish (Danio rerio) and human galectin IDRs. Source data are available online for this figure.

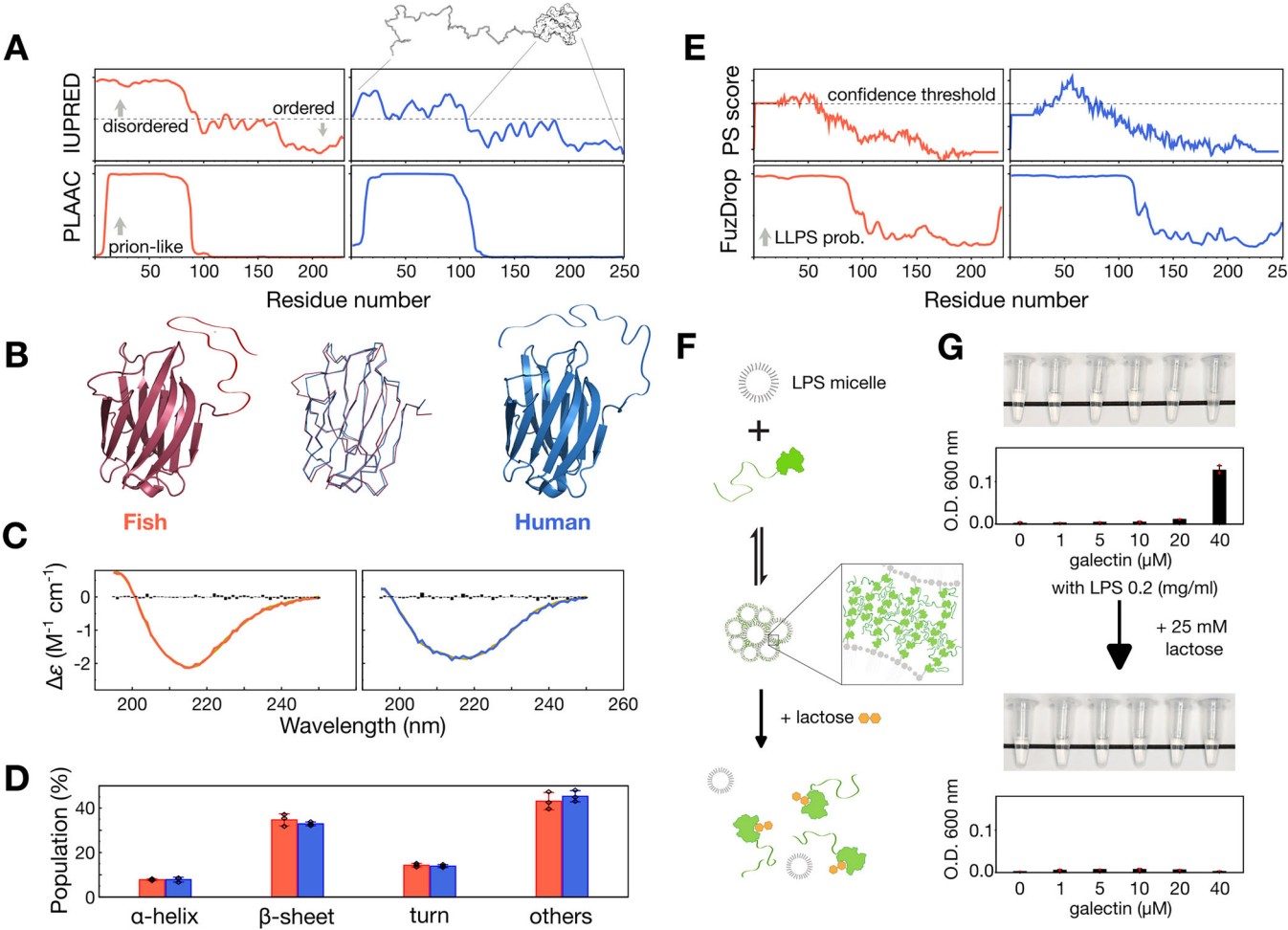

Figure 2.   Sequence and structural analysis of the IDR-tethered galectin in zebrafish.

(A) Levels of protein disorder and prion-likeness predicted respectively by IUPRED (Erdos et al, 2021) and PLAAC (Lancaster et al, 2014). (B) AlphaFold (Jumper et al, 2021) structural model of zebrafish galectin (left), experimentally determined structure of human galectin-3 (PDB: 2NMO; right), and best alignment of the two structures (middle). (C) Circular dichroism (CD) spectra of IDR-tethered zebrafish galectin and human galectin-3 (experimental data in red (zebrafish) or blue (human), best-fit curves from the BeStSel web server (Micsonai et al, 2015; Micsonai et al, 2018) in gray, and residuals as black bars). (D) Comparison of secondary structure components derived from BeStSel fitting. (E) Predicted phase separation propensities using PS score (Vernon et al, 2018) and FuzDrop web server (Hardenberg et al, 2020). (F) Schematic illustration of using lipopolysaccharide (LPS) micelles to test the agglutination capacity of galectins and their reversibility. (G) Photographs and corresponding optical density measurements at 600 nm of differently concentrated samples of zebrafish galectin mixed with 0.2 mg/ml LPS micelles (top) and after adding 25 mM lactose (bottom). (CD and optical density measurements were performed in triplicate and results are reported as mean ± s.d.). Source data are available online for this figure.

assays, based on our previous findings that LPS micelles are agglutinated by hGal3 and that the condensates can be reversed to monomers by adding lactose (a galectin-3's ligand that blocks interactions between the sugar moiety of the LPS micelles and the CRD, Fig. 2F) (Chiu et al, 2020). The fact that zebrafish galectin samples were turbid in the presence of LPS but became transparent again when lactose was added (Fig. 2G) suggests therefore that the IDR in zfGal drives agglutination as it in hGal3.

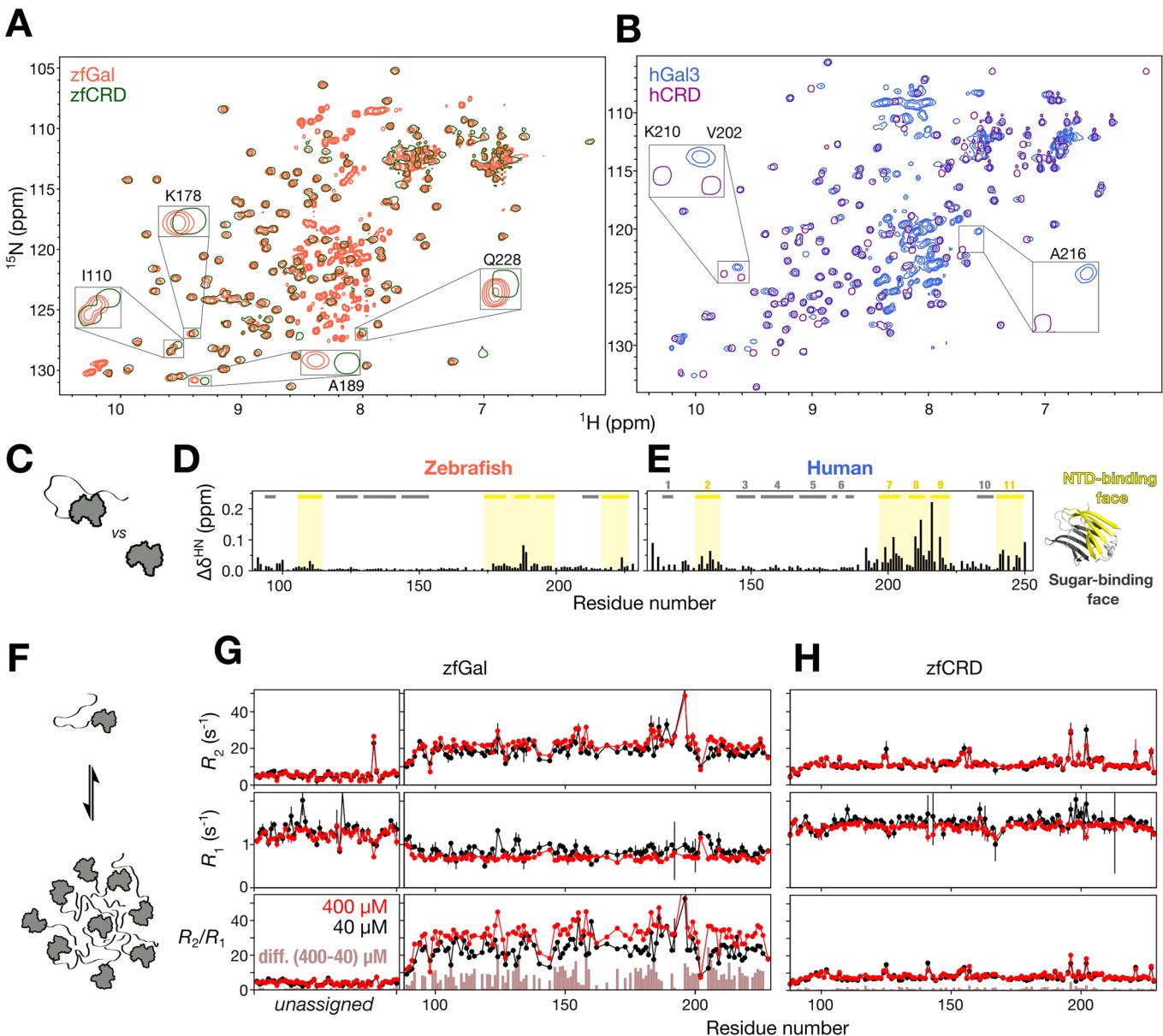

**Figure 3. Inter- and intramolecular interactions in zebrafish IDR-tethered galectin.**

(A, B) Overlaid HSQC spectra of full-length and carbohydrate-recognition domain (CRD)-only constructs of (A) zebrafish (full-length, orange; CRD-only, green) IDR-tethered galectin and (B) human (full-length, blue; CRD-only, purple) galectin-3. (C–E) Average chemical shift perturbations in the presence of the IDR for (D) zebrafish galectin and (E) human galectin-3 derived from the spectra shown in panels (A, B). Residues on the N-terminal domain binding side of the CRD are highlighted in yellow. (F–H) Transverse ($R_2$) and longitudinal ($R_1$) relaxation rate constants and $R_2/R_1$ ratio as a function of residue number (residues in the intrinsically disordered N-terminal domain are unassigned) measured in 400 μM (red) and 40 μM (black) samples of (G) full-length zebrafish galectin and (H) its CRD-only construct. Error bars indicate ±1 standard deviation (1σ) of the fitted $R_1$ and $R_2$ values estimated by Monte Carlo analysis (see Methods). The brown bars are the difference in $R_2/R_1$ ratios between high and low concentrations. Source data are available online for this figure.

## Similar inter- and intramolecular IDR–CRD interactions in human and zebrafish galectin

Human galectin-3 agglutinates via fuzzy inter- and intramolecular interactions between the intrinsically disordered NTD and CRD (Lin et al, 2017), which manifest in NMR chemical shift perturbations and relaxation rate changes. Although the CRD structures of zfGal and hGal3 are similar, their $^1H$-$^{15}N$

heteronuclear single-quantum coherence (HSQC) spectra are noticeably different (Fig. 3A,B; Appendix Fig. S2A,B). We thus assigned the chemical shifts of zfGal without the IDR (residues 1–83 removed, zfCRD; note that the IDR chemical shifts are uninformative for this analysis, which focuses on CRD sites that the IDR interacts with; Appendix Fig. S2C). The largest changes in peak positions between the CRD-only constructs and the full-length proteins are clustered in the non-carbohydrate-binding face,

**Table 1.** NMR dynamic analysis of zfGal protein constructs.

| Construct | zfGal | | zfCRD | | zfGal$^{W/Y}$ | |
|---|---|---|---|---|---|---|
| Conc. (µM) | 40 | 400 | 40 | 400 | 40 | 400 |
| $R_2$ | 17.03 ± 0.29 | 21.05 ± 0.31 | 10.05 ± 0.14 | 10.97 ± 0.17 | 18.83 ± 0.36 | 22.73 ± 0.49 |
| $|\Delta R_2|$ | 4.03 ± 0.43 | | 0.92 ± 0.22 | | 3.90 ± 0.61 | |
| $R_1$ | 0.82 ± 0.01 | 0.72 ± 0.01 | 1.48 ± 0.01 | 1.41 ± 0.01 | 0.76 ± 0.01 | 0.65 ± 0.01 |
| $|\Delta R_1|$ | 0.10 ± 0.01 | | 0.07 ± 0.01 | | 0.10 ± 0.01 | |
| $R_2/R_1$ | 20.65 ± 0.46 | 29.26 ± 0.54 | 6.79 ± 0.10 | 7.75 ± 0.13 | 24.90 ± 0.53 | 34.87 ± 0.84 |
| $|\Delta(R_2/R_1)|$ | 8.61 ± 0.71 | | 0.96 ± 0.16 | | 9.97 ± 1.00 | |

Dynamic analysis for the indicated constructs corresponding to Figs. 3G (zfGal), F (zfCRD) and 4F (zfGalW/Y). Only residues in the carbohydrate-recognition domain (CRD) were analyzed. For each condition (40 and 400 µM), $R_2$ and $R_1$ are inverse-variance weighted (IVW) means of the residue-wise estimates; standard errors (SEs) are the IVW SEs. $\Delta R_2$ and $\Delta R_1$ are the direct differences of the condition means. $R_2/R_1$ is the ratio of the IVW means at the same concentration, and $\Delta(R_2/R_1)$ is the difference between concentrations. Uncertainties for $R_2/R_1$, $\Delta R_2$, $\Delta R_1$, and $\Delta(R_2/R_1)$ are obtained by standard error propagation from the reported SEs of the IVW means.

indicating that these changes are due to the presence/absence of IDR interactions with the CRD (Fig. 3C,D). These chemical shift perturbations are less pronounced in zfGal (Fig. 3D) than they are in hGal3 (Fig. 3E).

To probe self-association, we compared concentration-dependent NMR spectra and dynamics to shift the monomer-oligomer equilibrium. Our earlier work showed that intermolecular interactions are negligible at 40 µM, whereas self-association is markedly increased at 400 µM without entering the PS regime (Lin et al, 2017). Consistent with this, the NMR spectra of zfGal at high (400 µM) and low (40 µM) concentrations overlap closely but the peak intensity ratios differ from the molar ratio (Appendix Fig. S3), most likely because of self-association (Fig. 3F). In NMR spin relaxation experiments, $R_2$ rates were higher and $R_1$ rates lower in the higher concentration sample (Fig. 3F,G; Table 1), which is typical of a shift in the equilibrium toward a bound (in this case self-associated) conformation (Fushman et al, 1997; Pfuhl et al, 1999; Korchuganov et al, 2001; Akerud et al, 2002; Baryshnikova and Sykes, 2006; Jensen et al, 2008). No such concentration-related differences were observed in the CRD-only construct (Fig. 3H; Table 1), indicating that this self-association is mediated by the intrinsically disordered NTD. Overall, therefore, these results suggest that intra- and intermolecular interactions in zfGal and hGal3 are similar but less pronounced in the former than in the latter.

## Effects of aromatic residue type on self-association and agglutination

To investigate the extent to which self-association and agglutination depend on the particular type of aromatic residue in the IDR, we replaced all the tryptophans in the IDR of zfGal with tyrosines, which predominate in hGal3 (Fig. 4A). Comparing the HSQC spectra of this zfGal$^{W/Y}$ mutant and of its CRD alone (Fig. 4B), the most pronounced chemical shift perturbations occur on the CRD's NTD-binding face (Fig. 4C). Furthermore, the peak intensities from residues on the NTD-binding face are less pronounced in the spectra of zfGal$^{W/Y}$ than they are in the spectrum of wild-type zfGal (Fig. 4D,E). These results indicate increased intramolecular interactions between the disordered and structured domains in zfGal$^{W/Y}$. Nevertheless, concentration-dependent relaxation rate profiles (Fig. 4F) suggest that zfGal$^{W/Y}$ has a similar propensity for

intermolecular self-association to wild-type zfGal's (see Fig. 3G; Table 1). The higher protein concentration of zfGal$^{W/Y}$ in the supernatant after LPS assays (Fig. 4G) suggests that the tyrosine-dominant construct in zfGal still has reduced agglutination capacities. Overall, therefore, these results suggest that the intermolecular self-association and agglutination capacity of IDR-bearing galectins is weakly dependent on the nature of aromatic residues in the IDR.

## Effect of the number of aromatic motifs on agglutination

Another difference between zfGal and hGal3 is the number of repeated aromatic-containing motifs in the IDR (and its overall length). To investigate the importance of this factor, we engineered an NTD-augmented construct of zebrafish galectin (zfGal$^{aug}$, Fig. 5A), with six extra tryptophan-containing tetrapeptides to match the length and number of aromatic residues in the IDR of hGal3 (Fig. 5B). Reversible agglutination in LPS micelles was still observed (Fig. 5C). The chemical shift differences between zfGal$^{aug}$ and the CRD alone suggest that intramolecular interactions are similar to those observed in wild-type zfGal (Fig. 5D), but less pronounced than those in zfGal$^{W/Y}$ (Fig. 4C). However, zfGal$^{aug}$ samples precipitated rapidly, and the low concentration (40 µM) NMR spectrum showed significant line broadening compared with that of wild-type zfGal (Appendix Fig. S4). Furthermore, while according to the length of their IDRs, the dynamic properties of zfGal3$^{aug}$ should be similar to those of hGal3 (ignoring self-association), the $R_2$ values measured at a concentration of 40 µM are higher than those measured in hGal3 at the same concentration (Fig. 5E), indicating a greater tendency toward intermolecular self-association. Meanwhile, LPS micelle assays indicated that zfGal$^{aug}$'s agglutination capacity is much higher than wild-type zfGal's (Fig. 5F; nearly no protein detected in the supernatant). The increased agglutination ability of zfGal$^{aug}$ relative to hGal3, despite their same IDR length and number of aromatic residues, is perhaps due to differences in the spacing of aromatic motifs in their IDRs that affect the strength of multivalent assembly (Martin et al, 2020; Holehouse et al, 2021) and solubility (Farag et al, 2022). These findings, therefore, indicate that the number of aromatic residues in the IDR is a more important determinant of galectin self-association and agglutination than the nature of these aromatic residues.

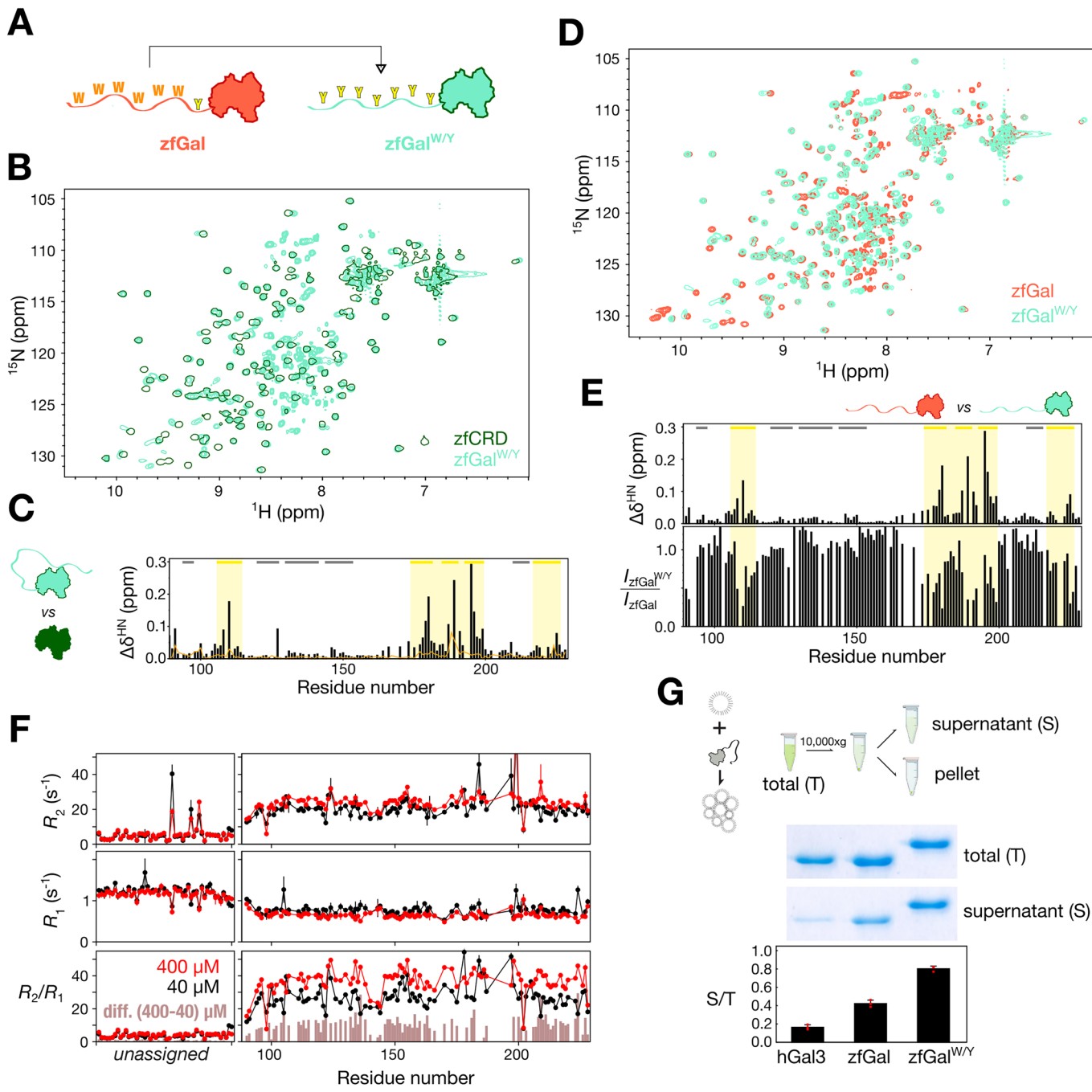

**Figure 4. Effects on self-assembly of aromatic residue type in zebrafish galectin.**

(A) Schematic illustration of the replacement of the tryptophans in zebrafish galectin's intrinsically disordered N-terminal domain (NTD) with tyrosines, yielding the zfGal$^{W/Y}$ construct. (B) Overlaid HSQC spectra of full-length (light green) zfGal$^{W/Y}$ and its carbohydrate-recognition domain (CRD) alone (dark green), and (C) the corresponding average chemical shift perturbations between the two (black bars). The orange line is the same comparison for wild-type zfGal (as shown in Fig. 3D). (D) Overlaid HSQC spectra of full-length zfGal$^{W/Y}$ (green) and wild-type zfGal (orange), and corresponding (E) average chemical shift differences and intensity ratios. Residues on the NTD interacting side of the CRD are highlighted in yellow. (F) Transverse ($R_2$) and longitudinal ($R_1$) relaxation rate constants and $R_2/R_1$ ratio as a function of residue number (the NTD is unassigned) measured in 400 µM (red) and 40 µM (black) samples of zfGal$^{W/Y}$. Error bars indicate ±1 standard deviation (1σ) of the fitted $R_1$ and $R_2$ values estimated by Monte Carlo analysis (see Methods). The brown bars are the difference in $R_2/R_1$ ratios between high and low concentrations. (G) Schematic illustration of lipopolysaccharide (LPS) agglutination assays. Representative SDS-PAGE analysis and protein amounts in the supernatant (S) of centrifuged LPS/protein mixtures determined by Bradford assays normalized to the total (T) amount of protein added for wild-type human galectin-3 (hGal3), zfGal$^{W/Y}$, and zfGal. Measurements were performed in triplicate, and results are reported as mean ± s.d. Source data are available online for this figure.

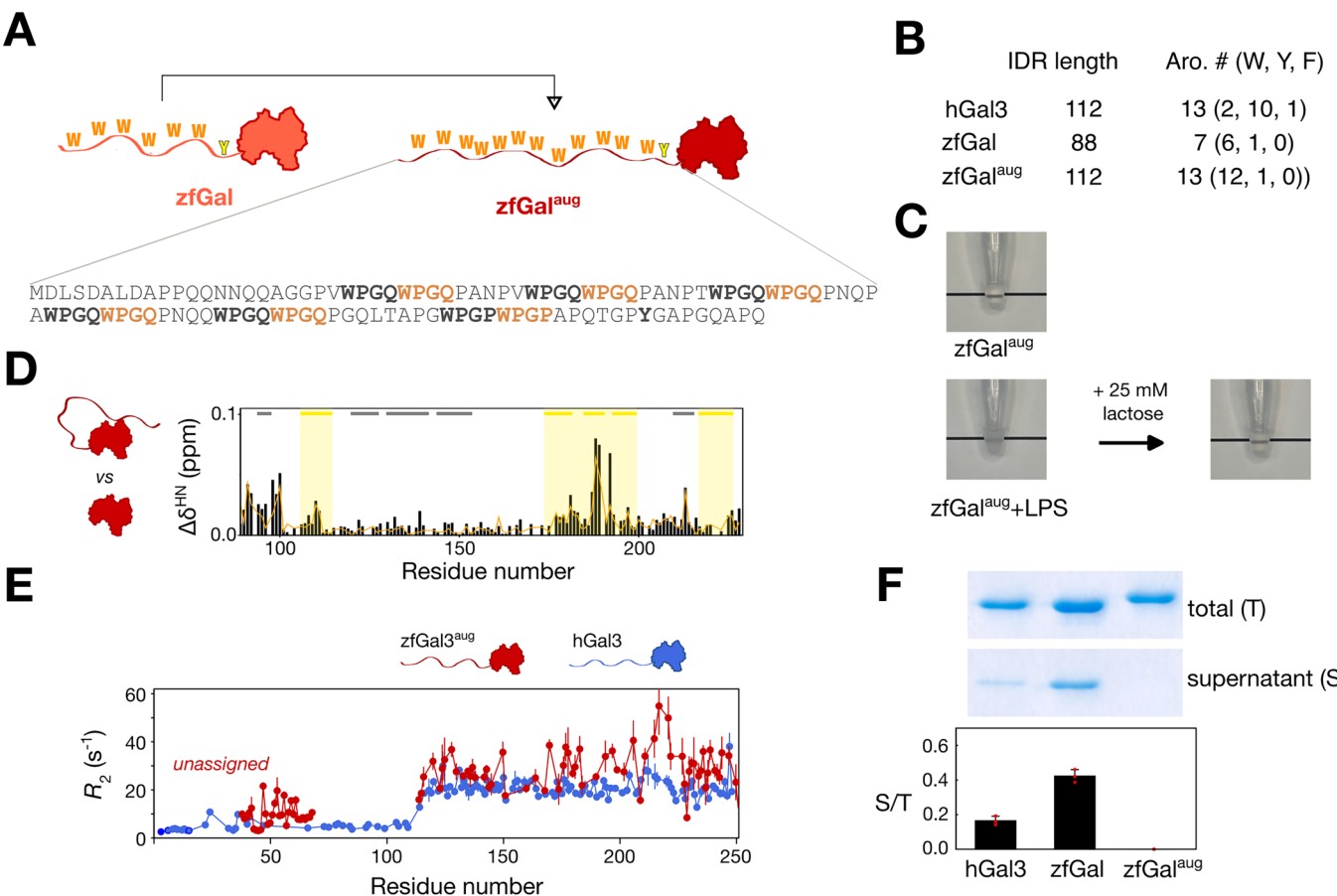

**Figure 5. Effects on self-assembly of the number of aromatic-containing motifs in zebrafish galectin.**

(A) Six tetrapeptides in the native sequence (bold font) of zebrafish galectin (zfGal) were duplicated to extend the intrinsically disordered region (IDR), yielding the zfGal^aug construct. (B) Comparison of the lengths and number of aromatic residues in the IDRs of human galectin-3 (hGal3), zfGal, and zfGal^aug. (C) Photographs of zfGal^aug samples becoming turbid after mixing with lipopolysaccharide (LPS) micelles (left) and becoming transparent once more after adding lactose (right). (D) Chemical shift differences between the presence and absence of the IDR of zfGal^aug. The orange line is the same comparison for wild-type zfGal (as shown in Fig. 3D). (E) Transverse relaxation rate constants ($R_2$) in 40 μM samples of zfGal^aug (red) and hGal3 (blue). (F) Representative SDS-PAGE analysis and protein amounts in the supernatant (S) of centrifuged LPS/protein mixtures determined by Bradford assays normalized to the total (T) amount of protein added for hGal3, zfGal (the same data as in Fig. 4G), and zfGal^aug. Measurements were performed at least in triplicate, and results are reported as mean ± s.d. Source data are available online for this figure.

## Contribution to self-association of the conserved N-terminal fragment

Apart from the repeated motifs on the same species, the motif analysis also revealed that the N-terminal residues of the IDRs are conserved across vertebrates (Fig. 1G). To investigate the role of this conserved fragment, we constructed a human CRD tethered to the first 21 residues of the NTD (a truncated construct without IDR residues 22–113; hGal3^Δ22–113). Although the agglutination ability of hGal3^Δ22–113 was similar to the CRD-only construct (Appendix Fig. S5), the $R_2$s of hGal3^Δ22–113 still increased between protein concentrations of 40 and 400 μM (Fig. 6A). This increase was greater than observed for the CRD alone (compare the brown bars $\Delta R_2$ in Fig. 6A,B). Per-residue difference-of-differences in transverse relaxation rate [$\Delta\Delta R_2 = \Delta R_2(\text{hGal3}^{\Delta 22-113}) - \Delta R_2(\text{hCRD})$] are mostly positive and a one-sided sign test rejects the null hypothesis that the median $\Delta\Delta R_2$ is zero (Fig. 6D). These analyses indicate that the observed $R_2$ enhancement in hGal3^Δ22–113 construct is a true effect and not attributable to random noise or variability, suggesting that the conserved fragment promotes self-assembly. Similar increases in $R_2$s were also

observed in a construct with IDR aromatic residues replaced by glycines (hGal3^WY/G; Fig. 6C), which has no agglutination capacity because the substituted aromatic residues provide the π-π and cation-π interactions required for PS (Chiu et al, 2020; Sun et al, 2024). The similar $R_2$ increments in these two constructs support the involvement of the conserved fragment in self-association. The HSQC spectra of hGal3^WY/G and hGal3^Δ22–113 show similar chemical shift perturbations relative to the CRD-only construct (Fig. 6D,E; Appendix Fig. S6), which likewise suggests that the interactions involve the N-terminal residues. The perturbations in these constructs are most pronounced in β-strands 2 and 11 (Fig. 6F), rather than in β-strands 7–9 for wild-type hGal3 (Fig. 3E). Contacts between this short fragment and these two β-strands may contribute to the observed weak self-association.

## Discussion

Sequence duplication is one of the most primordial modes of protein evolution (Lupas et al, 2001; Soding and Lupas, 2003; Kashi and King,

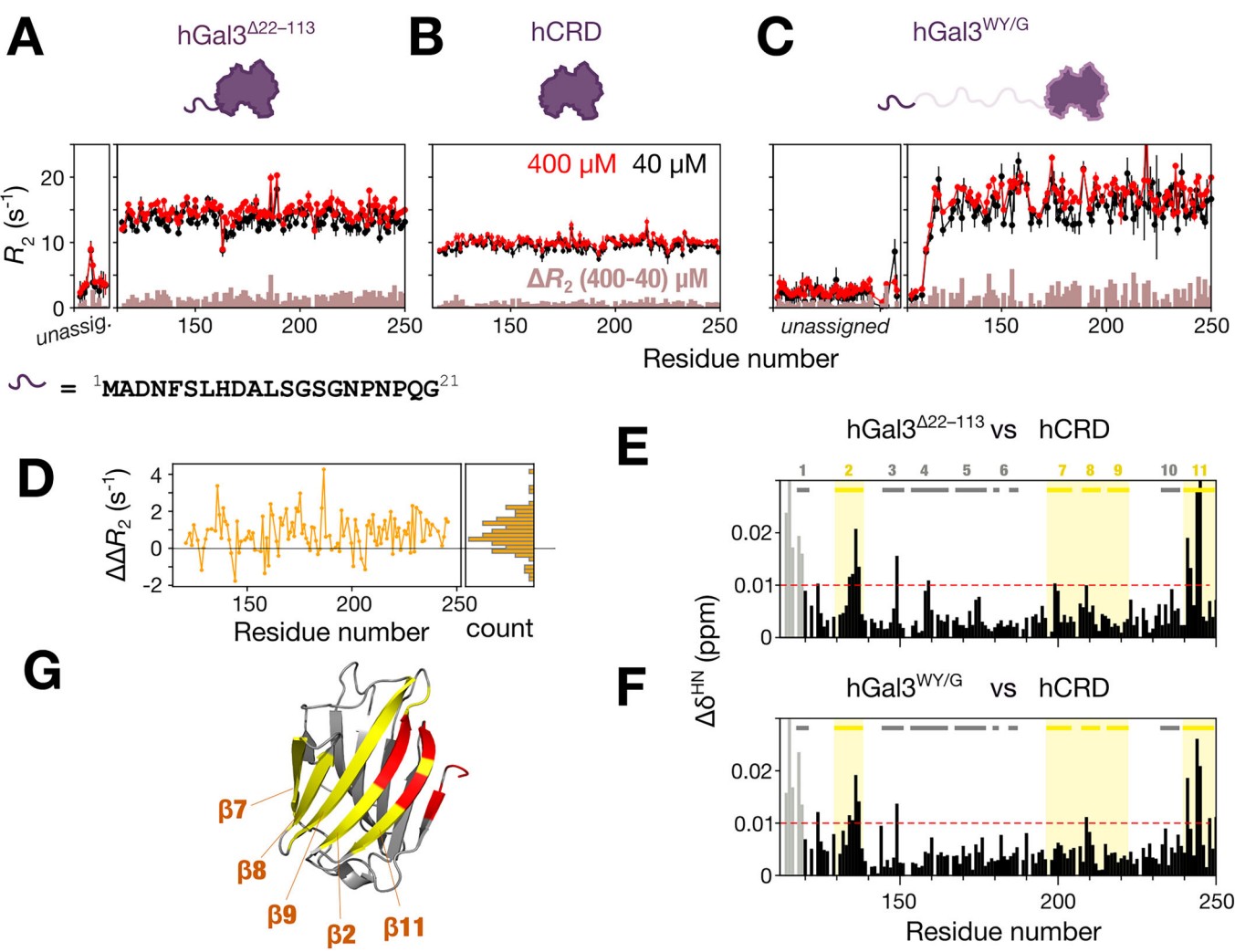

$\wedge\!\!\!\sim$ = $^1$MADNFSLHDALSGSGNPNPQG$^{21}$

**Figure 6.   Contribution to self-association of the conserved N-terminal motif in IDR-tethered galectins.**

(A–C) Transverse relaxation rate constants ($R_2$) of 400 µM (red) and 40 µM (black) samples of (A) human galectin-3 with only the first 21 residues of the IDR present (hGal3$^{\Delta22-113}$), (B) a structured domain-only (CRD-only) construct, and (C) a construct with tryptophans and tyrosines in the intrinsically disordered region replaced by glycine (hGal3$^{WY/G}$). Error bars indicate ±1 standard deviation (1σ) of the fitted $R_2$ values estimated by Monte Carlo analysis (see Methods). The differences between the two concentrations are shown as brown bars. (D) Per-residue difference-of-differences in transverse relaxation rate [$\Delta\Delta R_2 = \Delta R_2(hGal3^{\Delta22-113}) - \Delta R_2(hCRD)$]. The histogram (right panel) shows the distribution of $\Delta\Delta R_2$ values. A one-sided sign test rejects the null hypothesis of no shift ($p = 7.1 \times 10^{-12}$). The chemical shift perturbation (E) between hGal3$^{\Delta22-113}$ and the CRD-only construct and (F) between hGal3$^{WY/G}$ and the CRD-only construct. The perturbations in the N-terminus of the structured domain are due to sequence variations (gray bars). (G) Residues with chemical shift perturbations greater than 0.01 ppm (an arbitrary threshold for indicating the most pronounced perturbations) are highlighted in red on a structural model of the CRD. Source data are available online for this figure.

2006; Alva et al, 2015), as highlighted in the seminal work of Eck and Dayhoff (Eck and Dayhoff, 1966), which explains the emergence of a folded protein from repeated short motifs introduced via structure-promoting amino acids (e.g., cysteine or tryptophan). Large-scale gene duplications over many generations create paralogs in genomes, offering opportunities for new functions to evolve, leading over time to the development of highly specialized folded proteins (Ohno, 1970; Britten, 2006; Innan and Kondrashov, 2010). The evolution of protein does not stop at a structured form. As complicated cells, such as eukaryotic cells and multicellular organisms, have evolved, a protein could have "moonlight" (Jeffery, 1999) multiple functions, with the need for spatiotemporal controls for various functions. The emergence of IDRs with distinct physicochemical properties fulfills such

requirements, notably through PS (Pritisanac et al, 2020; Holehouse and Kragelund, 2024). While there may be many pathways for IDRs to evolve, motif duplication would represent a relatively straightforward route (Lupas et al, 2001; Soding and Lupas, 2003; Kashi and King, 2006; Alva et al, 2015), consistent with our observation for galectins.

Galectin-3's CRD is widely distributed in diverse organisms, as evidenced by the ~17,000 sequences in Fig. 1A and reference (Cooper and Barondes, 1999). Our analysis further reveals that these homologs can also evolve with IDRs (Fig. 1; Dataset EV2). Notably, these IDRs are not always tethered to the N-terminal region of a CRD. For instance, in one plant example, the IDR is tethered to the C-terminal region, whereas in nematode examples, IDRs are attached to tandem-repeated CRDs (Appendix Fig. S7;

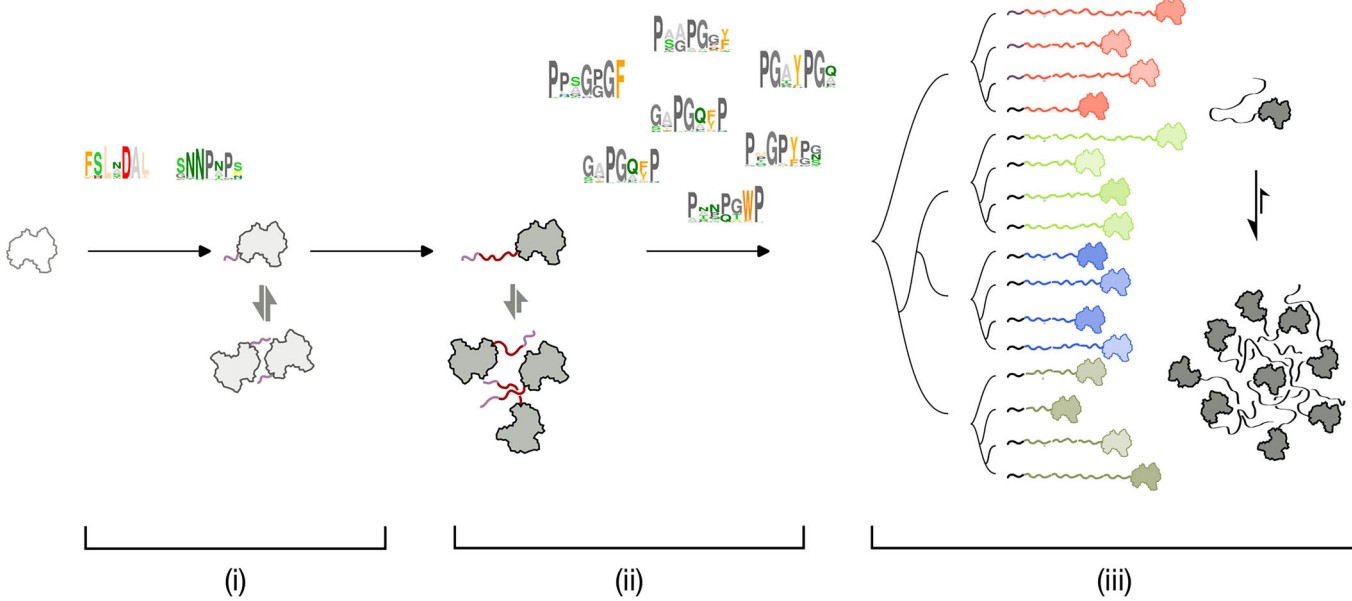

**Figure 7. Proposed evolutionary route for IDR-tethered galectin.**

(i) An ancestral galectin, harboring a single carbohydrate-recognition domain (CRD), acquires an incipient fragment (light purple) that facilitates self-association. (ii) An aromatic residue-containing motif (dark purple) inserted into this fragment, diversifying over time in both the aromatic residue types and the motif context. (iii) Different lineages fix the type(s) of aromatic residues in these motifs and then independently duplicate them, producing varying numbers of repeats. Natural selection fine-tunes the number of repeats for the fitness of assembly strength.

Dataset EV2). Their diverse amino acid compositions, variable lengths, and differing N-/C-terminal positions indicate that these IDRs have likely evolved independently across distinct lineages (Fig. 1B; Appendix Fig. S7).

Interestingly, while vertebrate's galectin IDRs share similar overall amino acid compositions, the specific types of aromatic residues differ (Fig. 1C). For example, the IDR-tethered galectin of zebrafish is rich in tryptophans rather than tyrosines in human's galectin-3 (Fig. 2). Our results indicate that the reduced self-assembly and agglutination ability of this zebrafish galectin (Fig. 3) is not due to this difference in amino acid type (Fig. 4), but rather to the number of repeated aromatic motifs in the amino acid sequence (Fig. 5). While recent studies have highlighted the role of aromatic residues in driving PS by acting as "stickers" (Wang et al, 2018; Bremer et al, 2022), our findings suggest that natural selection does not always favor a specific type of aromatic residue. One of the possible reasons may stem from the interactions between NTD and CRD. As demonstrated in our previous NMR analyses, positively charged residues on the CRD's non-ligand-binding surface (the NTD-binding face) contribute to overall assembly via cation-π interactions (Sun et al, 2024). Although the zebrafish CRD is conserved relative to the human ortholog, the number and spatial arrangement of these basic residues differ. Consequently, replacing tryptophans with tyrosines does not necessarily enhance assembly strength, as might be expected from model peptide studies. Instead, our data suggest that the evolutionary optimization of galectin assembly is achieved not by selecting the strongest individual "sticker" residue, but by duplicating the aromatic residue-containing motif multiple times, which seems to be the more efficient evolutionary strategy.

Different repeated motifs in IDRs may provide adaptive advantages. Fish, inhabiting environments spanning from cold to warm and from fresh to more or less salty, represent the most diverse class of

vertebrates. Although fish have undergone multiple gene duplication events during their evolution history (Meyer and Schartl, 1999), our analysis reveals that they also exhibit the greatest variation in galectin IDR lengths among vertebrates (Appendix Fig. S8). Furthermore, while the correlation between the number of aromatic residues in fish galectin IDRs and living environments is not clear, fish generally have more IDR-tethered galectins than other vertebrates (Appendix Fig. S8). It is plausible that galectins with various IDR lengths may have evolved in fish as an adaptation to environmental changes. For example, multiple copies of IDR-tethered galectins in trout and salmon with different numbers of aromatic motif repeats (Appendix Fig. S8) may have evolved as adaptations to their life cycle during which they transit between fresh, brackish, and salt water.

Human galectin-3, a β-galactoside-binding lectin, is unique among the galectin family for not being in dimer or tandem-repeat form (Yang et al, 2008). Phylogenetic analysis showed that a CRD-only structure as a common ancestor (Houzelstein et al, 2004) may have been duplicated in tandem-repeat form (e.g., human galectin-4 and -8) or have been selected through mutations that promote dimerization (e.g., human galectin-1 and -2). How, then, did galectin-3's unique chimeric form with a long-disordered region emerge? It is unlikely that the IDR originated through "destructuring" of a CRD because the IDR sequence differs substantially from other tandem-repeat galectins. Instead, the conserved N-terminal sequence found in vertebrate galectins, which promotes self-association (Fig. 6), may have seeded the eventual emergence of extended IDRs. This early conserved fragment might have played an evolutionary role analogous to that of the functional shift of a "2% wing" in the emergence of avian flight (Gould, 1985) or to the incidental assembly of an enzyme or chaperon in the developing lens of animal eyes (Shimeld et al, 2005; Piatigorsky, 2006). Over time, short aromatic-containing motifs were appended to

this incipient fragment to facilitate assembly. These motifs likely diversified and were independently duplicated in different species, ultimately creating varying numbers of repeats. Environmental pressures would then have fine-tuned these repeats to optimize agglutination strength in each species, thereby shaping the evolution of galectin IDRs in vertebrates (Fig. 7).

In this proteome-wide investigation focused on human galectin-3, instead of using homolog databases based on sequence or functional similarity, we specifically identified galectins with long IDRs and explored their evolutionary paths from fish to humans. Our findings suggest that the inherent sequence diversity of IDRs is crucial to their evolutionary adaptability, and this insight advances our understanding of the molecular basis of evolutionary biology, bringing our another "inner fish" (Shubin, 2008) to the fore.

# Methods

### Reagents and tools table

| Reagent/resource | Reference or source | Identifier or catalog number |
|---|---|---|
| **Chemicals, enzymes and other reagents** | | |
| Acetic acid | J.T.Baker | 9508 03 |
| Acetone | Honeywell | 32201 |
| Agar | BioShop Canada, Inc. | AGR001.1 |
| Agarose | UniRegion | UR-AGA001 |
| Ammonium chloride (NH$_4$Cl) | Sigma-Aldrich | 213330 500 G |
| Ammonium chloride ($^{15}$N, 99%) | Cambridge Isotope Laboratories, Inc. | NLM-467-1 |
| Ammonium persulfate (APS) | Sigma-Aldrich | A3678-25G |
| Ampicillin sodium salt | Gold Biotechnology, Inc. | A-301-100 |
| Bromophenol blue | BioShop Canada, Inc. | BRO777 |
| Basal Medium Eagle (BME) Vitamin Concentrate 100x Powder | United States Biological, Inc. | B0110 |
| Calcium chloride anhydrous | Sigma-Aldrich | C4901-100G |
| Coomassie Brilliant Blue G | Sigma-Aldrich | B-0770 |
| Coomassie brilliant blue G250 | Merck | 1.15444.0025 |
| cOmplete Protease Inhibitor Cocktail | Roche | 000000005892791001 |
| Deoxyribonuclease I (DNase I) | Bionovas | DRB001.100 |
| Deuterium oxide, 99.9 atom % D, contains 0.75 wt. % 3-(trimethylsilyl) propionic-*2,2,3,3-d$_4$*acid, sodium salt | Sigma-Aldrich | 293040-25 G |
| D-(+)-Glucose | Sigma-Aldrich | G5767-500G |
| D-Glucose (U-$^{13}$C6, 99%) | Cambridge Isotope Laboratories, Inc. | CLM-1396-1 |
| Ethanol | J.T.Baker | 8006-05 |

| Reagent/resource | Reference or source | Identifier or catalog number |
|---|---|---|
| Ethylenediaminetetraacetic acid (EDTA) | Sigma-Aldrich | E4884-100G |
| Glycerol | BioShopCanada, Inc. | GLY001.1 |
| Glycine | J.T.Baker | 4059-06 |
| Imidazole | Alfa Aesar | A10221 |
| Isopropanol | J.T.Baker | 9084-03 |
| Isopropyl β-D-1-thiogalactopyranoside (IPTG) | Gold Biotechnology, Inc. | I2481C100 |
| Lactose | Sigma-Aldrich | 5989-81-1 |
| Lipopolysaccharide from *E. coli* serotype O55:B5 (LPS) | Sigma-Aldrich | SI-L2880-10MG |
| Lysozyme | BIONOVAS | AL0680-0010 |
| Magnesium chloride anhydrous | Sigma-Aldrich | M-8266-100G |
| Magnesium sulfate heptahydrate | Sigma-Aldrich | 230391-500 G |
| Methanol | Macron Fine Chemicals | 3016.68 |
| Phenylmethylsulfonyl fluoride (PMSF) | Sigma-Aldrich | P7626-5G |
| Potassium chloride | Sigma-Aldrich | P3911-500G |
| Potassium phosphate monobasic | Sigma-Aldrich | P5379-500G |
| RNase A | Sigma-Aldrich | R5503-100MG |
| Sodium chloride | Sigma-Aldrich | 31434-5KG-R |
| Sodium chloride | VWR Life Science | 0241-1KG |
| Sodium hydroxide | Sigma-Aldrich | 30620 |
| Sodium phosphate dibasic dihydrate | Sigma-Aldrich | 04272-1KG |
| Sodium phosphate monobasic | Sigma-Aldrich | S0751-500G |
| Sodium dodecyl sulfate (SDS) | Sigma-Aldrich | 75746-1KG |
| Tris hydroxymethyl aminomethane (Tris-base) | Bioman | TRS011 |
| Tryptone | Bioshop Canada, Inc. | TRP402.1 |
| Yeast extract | Bioshop Canada, Inc | YEX401.500 |
| Mini Plus Plasmid DNA Extraction System | Viogene | GF2001 |
| Gel/PCR DNA Isolation System | Viogene | GP1001 |
| 100 bp DNA Ladder | Bioman | DL100 |
| 1 kb DNA Ladder | Bioman | DL1000 |
| Gel Loading Dye (6X) | New England Biolabs, Inc. | B7024S |
| Acrylamide/Bis Acrylamide (29:1) 30% Solution | BioShopCanada, Inc. | ACR009.500 |
| DNA View | BioTools | TT-DNA01 |
| Prestained Protein Molecular Weight Marker | Bioman | Prep1025 |
| *DpnI* | New England Biolabs, Inc. | R0176S |

| Reagent/resource | Reference or source | Identifier or catalog number |
|---|---|---|
| Q5 High-Fidelity 2X Master Mix | New England Biolabs, Inc. | M0492S |
| T4 DNA Ligase | New England Biolabs, Inc. | M0202S |
| **Bacterial Strains** | | |
| ECOSTM 101 *E. coli* Competent Cells [DH5α] | Yeastern Biotech Co.,Ltd. | LYE678-80VL |
| ECOSTM 21 *E. coli* Competent Cells [BL21 (DE3)] | Yeastern Biotech Co.,Ltd. | FYE207-40VL |
| Rosetta™ (DE3) Competent Cells-Novagen | Sigma-Aldrich | 70954 |
| **Recombinant DNA** | | |
| His$_6$-SUMO-hGal3 | Lin et al, 2017 | N/A |
| His$_6$-SUMO-hGal3$^{WY/G}$ | Chiu et al, 2020 | N/A |
| His$_6$-SUMO-hGal3$^{Y/W}$ | Gene synthesis (Synbio) | N/A |
| His$_6$-SUMO-hGal3$^{Δ22-113}$ | This work | N/A |
| His$_6$-SUMO-hGal3$^{Δ20}$ | Lin et al, 2017 | N/A |
| His$_6$-SUMO-zfGal3 | Gene synthesis (Gene Script Inc.) | N/A |
| His$_6$-SUMO-zfCRD | This work | N/A |
| His$_6$-SUMO-zfGal3$^{W/Y}$ | Gene synthesis (Gene Script Inc.) | N/A |
| His$_6$-SUMO-zfGal3$^{aug}$ | Gene synthesis (Gene Script Inc.) | N/A |
| **Software and algorithms** | | |
| BeStSel | https://bestsel.elte.hu/information.php | |
| AlphaFold | https://alphafoldserver.com/ | |
| PLAAC | http://plaac.wi.mit.edu/ | |
| PONDR | http://www.pondr.com/ | |
| PScore | http://abragam.med.utoronto.ca/~JFKlab/Software/psp.htm | |
| FuzDrop | https://fuzdrop.bio.unipd.it/predictor | |
| ProtParam | https://web.expasy.org/protparam/ | |
| NMRPipe | https://www.ibbr.umd.edu/nmrpipe/ | |
| Sparky | https://nmrfam.wisc.edu/nmrfam-sparky-distribution/ | |
| I-PINE | http://i-pine.nmrfam.wisc.edu/ | |
| NMRtist | https://nmrtist.org/ | |
| PyMOL | https://pymol.org/ | |
| **Table of primer** | | |
| Construct name | Template | primer |
| His-SUMO-zfCRD | His-SUMO -zfGal3 | F: 5′ attggcggcggacaagctccacaagtg 3′<br>R: 5′ agcttgtccgccgccaatctgttctct 3′ |
| His-SUMO-hGal3$^{Δ22-113}$ | His-SUMO-hGal3 | F: 5′ caggaccactgattgtgccttataacctg 3′<br>R: 5′ cagtggtccttgagggtttgggtttccaga 3′ |

## Bioinformatics and sequence analysis

Protein sequence databases were obtained from the trEMBL database (accessed February 2023) (UniProt 2023). The BLAST (Basic Local Alignment Search Tool) (Zaru et al, 2023) algorithm was run locally to identify similar sequences. IUPRED3 (Erdos et al, 2021) was integrated with in-house written Python scripts designed to select proteins based on the length of their disordered regions. The species of the selected protein was retrieved from the corresponding FASTA file, and the phylogenetic relations were generated using the ete3 toolkit (Huerta-Cepas et al, 2016). MEME suite (motif-based sequence analysis tools) (Bailey et al, 2015) was applied to search sequence motifs.

Levels of structural disorder and prion-likeness were respectively estimated with the PONDR (Obradovic et al, 2005) and PLAAC (Lancaster et al, 2014) webservers. π-π interactions and PS were predicted using the PScore predictor (Vernon et al, 2018) and the FuzDrop server (Hardenberg et al, 2020). The structure of the CRD of IDR-tethered zebrafish galectin was predicted using AlphaFold (Jumper et al, 2021), accessed through the ColabFold interface (Mirdita et al, 2022).

## DNA constructs

The construction of plasmids for full-length, CRD-only, and aromatic residues-removed (WY/G) human galectin-3 has been described previously (Lin et al, 2017; Chiu et al, 2020). The construct with only the 21 N-terminal residues of the IDR present (hGal3$^{Δ22-113}$) was designed using the FastCloning (Li et al, 2011) method. The cDNA sequences of IDR-tethered zebrafish galectin (zfGal, zfGal$^{W/Y}$, and zfGal$^{aug}$) were created by whole gene synthesis (Gene Script Inc. and Synbio). A hexahistidine-tagged small ubiquitin-like modifier protein (His$_6$-SUMO) was used in each case to assist purification. All constructs were confirmed by sequencing.

## Protein purification and NMR data collection

All variants used in this study were purified as previously described (Lin et al, 2017; Chiu et al, 2020; Sun et al, 2024). NMR data, including HSQC spectral intensity, chemical shift analysis, chemical shift assignment, and transverse ($R_2$) and longitudinal ($R_1$) relaxation rate constants, were collected and analyzed following established protocols (Lin et al, 2017; Chiu et al, 2020; Sun et al, 2024). Detailed procedures are provided in the Appendix Methods.

## Circular dichroism analysis

Circular dichroism (CD) spectra were recorded on an AVIV model 410 spectropolarimeter at 303 K. Samples were loaded into a 0.1 mm cuvette. Ten measurements were co-added for each data point, between 190 and 260 nm, with an interval of 1 nm. The CD signal was normalized to the sample concentration and the number of residues. The measured theta machine units (θ) were converted to Δε using the following equation:(Greenfield, 2006)

$$\triangle\varepsilon = \theta \cdot \frac{0.1 \cdot MRW}{l \cdot C \cdot 3298}$$

where $l$ is the path length (in cm), $C$ is the protein concentration (in mg/ml), and MRW is the mean residue weight (molecular weight/residue number, in Dalton). Secondary structure populations were estimated using the BeStSel program (Micsonai et al, 2015; Micsonai et al, 2018). All measurements were conducted in three independent replicates.

## Lipopolysaccharide (LPS) agglutination assays

Lipopolysaccharide micelles from *E. coli* strain O55:B5 were obtained from Merck (Catalog No. L2880). A total amount of 20 μM protein (T) was mixed with a final concentration of 0.2 mg/ml LPS, which was then centrifuged at 10,000×*g* for 5 min to obtain the supernatant (S) and pellet. Protein concentrations (S, T) were determined using Bradford assays. Bovine serum albumin (BSA) standards and the assay samples were mixed with 170 μl of Bradford reagent, and the absorbance at 594 and 466 nm (A594 and A466) was measured using a TECAN Spark microplate reader. Sample concentrations were obtained from a linear regression of BSA concentrations as a function of A594/A466 ratios (Zor and Selinger, 1996). SDS-PAGE analyses were also performed to confirm the amount and integrity of the samples. All measurements were performed at least in triplicate.

## Data availability

All data and codes/scripts for analysis were deposited as the Source Data with this article online. The chemical shifts reported in this article are deposited in the Biological Magnetic Resonance Bank (BMRB) with the access number 52443. (https://bmrb.io/data_library/summary/index.php?bmrbId=52443).

The source data of this paper are collected in the following database record: biostudies:S-SCDT-10_1038-S44319-026-00692-w.

## Peer review information

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

## Acknowledgements

This work was supported by the National Science and Technology Council of Taiwan (110-2113-M-A49A-504-MY3 and 113-2113-M-A49-031-MY3), Yen Tjing Ling Medical Foundation (CI-110-16 and CI-111-19), the Higher Education Sprout Project by the Ministry of Education (MOE) in Taiwan. Y-HL and Y-CS are the recipients of the Taiwan International Graduate Program (TIGP) Rising Star Fellowship. We thank Profs. Jun-Yi Leu and Yung-Che Tseng (Academia Sinica) for their insightful discussion. The authors also thank the Academia Sinica High-Field NMR Center (HFNMRC) for technical support. (HFNMRC is funded by Academia Sinica Core Facility and Innovative Instrument Project (AS-CFII-111-214).

## Author contributions

**Yu-Hao Lin**: Conceptualization; Data curation; Formal analysis; Supervision; Investigation; Visualization; Writing—original draft. **Yu-Chen Chen**: Data curation; Formal analysis; Investigation; Visualization; Methodology; Writing—original draft. **Yung-Chen Sun**: Investigation. **Jie-rong Huang**: Conceptualization; Resources; Data curation; Software; Formal analysis; Supervision; Funding acquisition; Validation; Investigation; Visualization; Methodology; Writing—review and editing.

Source data underlying figure panels in this paper may have individual authorship assigned. Where available, figure panel/source data authorship is listed in the following database record: biostudies:S-SCDT-10_1038-S44319-026-00692-w.

## Disclosure and competing interests statement

The authors declare no competing interests.

# Expanded View Figures

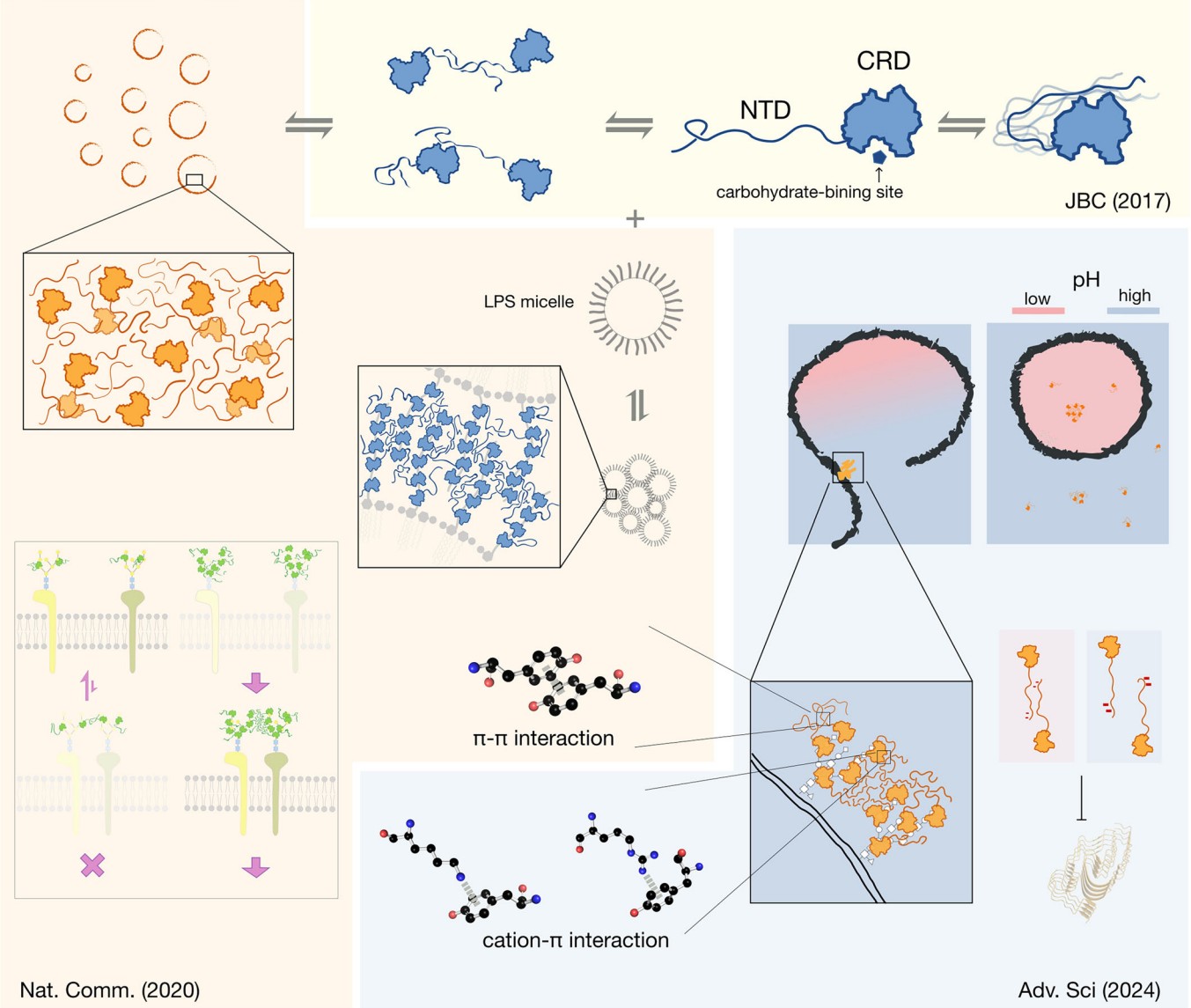

**Figure EV1. Summary of previous studies on galectin-3 self-association.**

Background shading groups panels by publication: yellow: JBC (Lin et al, 2017); peach: Nat. Commun. (Chiu et al, 2020); blue: Adv. Sci. (Sun et al, 2024). Galectin-3's intrinsically disordered N-terminal domain (NTD) forms transient intra- and intermolecular contacts with the non-carbohydrate-binding face of its carbohydrate-recognition domain (CRD; the NTD-binding face) and also self-associates (yellow). These interactions drive phase separation (PS) and enable galectin-3-mediated agglutination, assessed with lipopolysaccharide (LPS) micelles. Agglutination is supported by π-π interactions among aromatic residues within the NTD (peach) and by cation-π interactions between these aromatics and conserved, positively charged residues on the NTD-binding face. Two conserved acidic residues in the NTD modulate self-association in a pH-dependent manner (blue).

