## [Peer Review File · EMBO Reports]

Evolution of intrinsically disordered regions in vertebrate galectins for phase separation

Yu-Hao Lin, Yu-Chen Chen, Yung-Chen Sun, and Jie-rong Huang

Corresponding author(s): Jie-rong Huang (jierongh@nycu.edu.tw)

Review Timeline:

Submission Date:	23rd Jun 25
Editorial Decision:	6th Aug 25
Revision Received:	27th Oct 25
Editorial Decision:	15th Dec 25
Revision Received:	16th Dec 25
Accepted:	12th Jan 26

Editor: Yehu Moran

Transaction Report:

Dear Dr. Huang

Thank you for the submission of your manuscript to EMBO Reports. We have now received the full set of referee reports that are all pasted below.

As you will see, the referees acknowledge that the findings are interesting and relevant for the community. However, they also raise a considerable number of concerns and comments that require your attention.

I would thus like to invite you to revise your manuscript with the understanding that the referee concerns must be fully addressed and their suggestions taken on board. Please address all referee concerns in a complete point-by-point response. Acceptance of the manuscript will depend on a positive outcome of a second round of review. It is EMBO Reports policy to allow a single round of major revision only and acceptance or rejection of the manuscript will therefore depend on the completeness of your responses included in the next, final version of the manuscript.

We realize that it is difficult to revise to a specific deadline. In the interest of protecting the conceptual advance provided by the work, we recommend a revision within 3 months (6th Nov 2025). Please discuss the revision progress ahead of this time with the editor if you require more time to complete the revisions.

- 1) A data availability section providing access to data deposited in public databases is missing. If you have not deposited any data, please add a sentence to the data availability section that explains that.
- 2) Your manuscript contains statistics and error bars based on $n=2$. Please use scatter blots in these cases. No statistics should be calculated if $n=2$.

<<https://www.embopress.org/page/journal/14693178/authorguide#expandedview>>

5) a complete author checklist, which you can download from our author guidelines

<<https://www.embopress.org/page/journal/14693178/authorguide>>. Please insert information in the checklist that is also reflected in the manuscript. The completed author checklist will also be part of the RPF.

6) Please note that all corresponding authors are required to supply an ORCID ID for their name upon submission of a revised manuscript (<<https://orcid.org/>>). Please find instructions on how to link your ORCID ID to your account in our manuscript tracking system in our Author guidelines <<https://www.embopress.org/page/journal/14693178/authorguide#authorshipguidelines>>

7) Before submitting your revision, primary datasets produced in this study need to be deposited in an appropriate public database (see <https://www.embopress.org/page/journal/14693178/authorguide#datadeposition>). Please remember to provide a reviewer password if the datasets are not yet public. The accession numbers and database should be listed in a formal "Data Availability" section placed after Materials & Method (see also <https://www.embopress.org/page/journal/14693178/authorguide#datadeposition>). Please note that the Data Availability Section is restricted to new primary data that are part of this study. * Note - All links should resolve to a page where the data can be accessed. *
If your study has not produced novel datasets, please mention this fact in the Data Availability Section.

12) All Materials and Methods need to be described in the main text using our 'Structured Methods' format, which is required for all research articles. According to this format, the Methods section includes a Reagents and Tools Table (listing key reagents, experimental models, software and relevant equipment and including their sources and relevant identifiers) followed by a Methods and Protocols section describing the methods using a step-by-step protocol format. The aim is to facilitate adoption of the methodologies across labs. More information on how to adhere to this format as well as a downloadable template (.docx) for the Reagents and Tools Table can be found in our author guidelines: <https://www.embopress.org/page/journal/14693178/authorguide#structuredmethods>.

An example of a Method paper with Structured Methods can be found here: <https://www.embopress.org/doi/full/10.1038/s44320-024-00037-6#sec-4>

I look forward to seeing a revised form of your manuscript when it is ready.

Yours sincerely,

Yehu Moran
Academic Editor
EMBO Reports

Referee #1:

Summary

The manuscript "Evolution of intrinsically disordered regions in vertebrate galectins for phase separation" analyses how the IDR of hGal3 has evolved through vertebrate lineages. The authors find that all the vertebrate lineages show repeating motifs with similar sequence patterning that always includes an aromatic residue, and while the residue identity is conserved within the repeats, it is not between different lineages, indicating that the aromatic residue identity diverged prior to the repeating motifs expanding. They find that the aromatic residue is required for function, however the identity of the residue is not critical. They also find a region at the N-terminus of the IDR that is conserved across vertebrates that contributes to self-association by interacting with the folded CRD.

The work is interesting and should appeal to an increasingly broad audience of people interested in IDRs and how they mediate phase separation.

Major comments/questions:

The authors show that you can get similar behavior between the Zebrafish and human orthologs even though the Zebrafish has Tryptophan residues in its repeats while the human has Tyrosine. Oddly they find that substituting Tyrosine in place of Trp on the Zebrafish resulted in increased interactions between the CRD and NTD. This is interesting as Tryptophans usually have a stronger interaction strength than Tyrosine, as seen in their own zfGal3 variant as well as in the literature (see 10.1091/mbc.E24-03-0128 and 10.1038/s41567-024-02558-1). Do the authors have any explanation for this?

For Figure 6 (primarily, but this affects the other chemical shift plots as well) they show a line at 0.01PPM and treat shifts greater than this as significant, but they do not directly state that this is the digital resolution of the experiments. Furthermore, they do not give the number of increments for the experiments in the methods, so it is impossible of the reader to tell if this is significant or not. Additionally, the difference in relaxation rates between the 40 and 400uM measurements is quite close to the noise in the experiment (See 6A, and 6C). The case for this region being important would be made stronger if there was complimentary evidence. Perhaps NMR in Trans, (i.e. 40uM of CRD plus 400uM of the Nterm residues?). Or the solubility experiments in 4G, and 5F with a construct lacking this Nterm domain. Perhaps even adding the AlphaFold3 prediction showin an interaction between this N-terminal region and the CRD as well.

But if this is true, how does this affect the model? If there is a specific interaction between this N-terminal region and the CRD, could it not act as an autoinhibitory element, a common feature in tethered IDRs? Given that they are tethered to one another the effective concentration would be quite high (likely in the mM range) perhaps increasing the concentration required for higher order assemblies/phase separation to occur. This could perhaps also explain the increased intramolecular interactions found in the zfGalW/Y variant. For literature on the effect of linker length on self/autoinhibitory-interactions there are extensive works by Magnus Kjaergaard and Lucia Chemes.

Minor Comments/Questions:

There is still an ongoing thread in the literature suggesting that hGal3 functions as a dimer. The authors addressed/refuted this in their earlier work, but perhaps it is still necessary to repeat that.

Figure 1.

1E, can the authors provide a legend of what the different colored modules in the IDR denote.

Figure 2/7.

2F, 7ii. Model of galectin interactions leading to PS, it looks like the folded domain/NTD, is main only interaction, shouldn't there also be some, or perhaps better indication that the aromatics within the IDR are adding to the interactions.

Referee #2:

The manuscript entitled "Evolution of intrinsically disordered regions in vertebrate galectins for phase separation" by Chen et al. presents a deep bioinformatics and structural investigation of several Galectin-3 forms to decipher the role of the composition of the IDR in intramolecular interactions, self-association and phase separation. The authors show that the IDR is responsible of these functions regardless of the specific aromatic amino acids present in these tails. Using NMR they unambiguously show that tryptophan and tyrosine, although with different intensity, play a fundamental role in these functions. Indeed, Modulation of the self-interaction and agglutination are mainly dictated by the number of aromatic-containing repeats more than the amino acid. Finally, the authors identify a highly conserved segment at the N-terminus of the IDR that plays a fundamental role in self-association.

The study is very elegant and properly done and results are clearly described. Conclusions are solid and in accordance with the results obtained. In my opinion, the authors convincingly disentangle the functional role of aromatics in galectin-3, and present a model that can be probably applied to many other protein families.

In my opinion, the functional role of the conserved N-terminal segment in self-association and intra-molecular interactions would be reinforced if a mutant without this fragment would be studied. This would reinforce the image (maybe simplistic) that this region is the only responsible of the intramolecular interactions.

Referee #3:

Comments for EMBOR-2025-62185V1

Title: Evolution of intrinsically disordered regions in vertebrate galectins for phase separation

This manuscript provides a proteome-wide analysis of the evolution of intrinsically disordered regions (IDRs) in the galectin-3 family, investigating their roles in driving phase separation (PS) and agglutination. Building on their 2020 Nature Communications study-where they demonstrated that human galectin-3 (hGal3) uses PS to agglutinate LPS micelles-the authors now extend their investigation to zebrafish galectin (zfGal). They integrate sequence mining of over 1,300 galectin-3 homologs, computational predictions (disorder propensity, PS scoring, motif enrichment, π - π interaction potential), and targeted biophysical experiments (NMR HSQC and R_1/R_2 relaxation, circular dichroism, and LPS-based turbidity assays). By systematically comparing zfGal and hGal3-and swapping aromatic motifs and numbers-they dissect the relative contributions of motif number versus aromatic residue identity to agglutination, concluding greater importance for the latter feature.

Overall, the manuscript is interesting and worthy of publication in EMBO Reports. There are several experiments and revisions the authors could do to improve the manuscript, which I list below:

Major Comments (roughly in order of importance)

1. Confusing Logic

I found the logic of the experiments and the choices of comparisons hard to follow, making the authors' conclusions difficult to understand. Often apples are compared to oranges (e.g. R2 of hGal3 to zfGal3aug, which are hard to relate due to differences in aromatic residue type, IDR patterning and the CRD sequences), and apples to apples comparisons that should have been shown are not (e.g. R2 of zfGal3 WT to W/Y to aug, which show clear trends of self-association that parallel agglutination trends but are shown in different figures; also R2 of hGal3 vs hGal3WY/G, which also show patterns that parallel agglutination but are also shown in different figures). I would suggest a rearrangement of some of the data to make the relationships between sequence, R2 and agglutination easier to understand. Relatedly, a summary table of average R2 values for the CRD residues of the different proteins would facilitate comparison.

2. Missing Reciprocal Mutants

The authors swap $W \rightarrow Y$ in zfGal to mimic hGal3, but do not analyze the reverse ($Y \rightarrow W$) in hGal3 and compare it to zfGalaug. Nor do they make a zfGalaug-W/Y mutant to compare with zfGalaug. Testing these mutants would better confirm whether residue identity truly has minimal impact and are key symmetry checks.

3. Intra- vs intermolecular interactions

It is unclear how the authors know that the chemical shift perturbations in the various figures result from intramolecular interactions. No KD value is stated for the interactions, and even at 40 μ M there could be intermolecular interactions that perturb chemical shift. The authors should examine interactions between the IDR and CRD in trans to assess this possibility, and/or measure the relevant KD values.

4. Interactions of the N-terminal Motif

The conclusion from figure 6 is confusing compared to earlier data. In figure 6, chemical shift perturbations of the CRD are attributed to interactions with the N-terminal 1-21 motif. This motif does not have either W or Y residues, but earlier data show

that mutating W to Y in zfGal3 changes CRD chemical shift perturbations due to the IDR. How to reconcile the two observations?

5. Lack of Direct Condensate Imaging

The authors do not show DIC/fluorescence (or FRAP) images of zfGal or zfGalW/Y (+ LPS) droplets-only turbidity and NMR. They should include images (and FRAP kinetics if possible), ideally alongside their own hGal3 images from 2020, to unambiguously demonstrate liquid like phase separation of the zebrafish proteins.

6. Quantitative π - π Scores & Aromatic Composition

The authors quote PScore/FuzDrop but do not summarize those interaction scores or % of F/Y/W across their 1,321 hits or filtered 351 "hGal3 like" set. A heatmap or boxplot of π - π scores and aromatic percentages would help evaluate their evolutionary claims.

Minor Comments

1. No Color Legends in Key Figures

Panels like Fig 3A/B, 3G/H, 4E/F lack legends. The authors should consider putting label legends "black = 40 μ M; red = 400 μ M" and "green = CRD-only; blue = full-length" in the figures in addition to captions for visual ease of the reader.

2. No Direct hGal3 vs. zfGalW/Y Visuals in Fig 4

Per major point 1 above, the authors present zfGalW/Y data but never juxtapose it with hGal3. A side by side panel for HSQC overlays and LPS assays would highlight the modest effect of aromatic type.

3. Undefined "Weak" vs. "Strong" Effects

The authors call aromatic-type effects "weak," but never set thresholds. They should define, e.g., $\Delta\delta < 0.05$ ppm as weak or specify a % change in R_2 that counts as significant (relative to errors in R_2).

4. No Statistics for R_2 Increase in $\Delta 22-113$

The $\Delta 22-113$ construct shows a small R_2 bump vs. CRD alone but lacks error bars or significance tests. A simple mean {plus minus} SD or t test would clarify if this is a real effect.

5. Overstated "Motif Duplication" Mechanism

The authors' claims that they "demonstrate" evolution via motif duplication is too strong based on a few example species. They need broader phylogenetic statistics on motif copy number changes or should soften wording to "consistent with" duplication as one plausible route.

6. Delineation of interaction types

The authors should do more in their analysis to delineate the roles of IDR-IDR and IDR-CRD interactions in promoting agglutination. E.g. in the zfGalW/Y mutant, IDR-CRD interactions increase, but agglutination decreases, in apparent contradiction (Fig. 4). This contradiction is resolvable, though, if one assumes that the IDR-IDR interactions would be weakened by the WY mutations, as seen with other phase separating systems. It is possible that the overall weak dependence of agglutination on aromatic residue type reflects trade offs between IDR-IDR and IDR-CRD interactions. This is worth discussing. In general, the authors should do more to distinguish the different types of interactions reported on in their data. Intermolecular vs intramolecular, IDR-IDR vs IDR-CRD, self-assembly vs agglutination.

Referee #1:

Summary

The manuscript "Evolution of intrinsically disordered regions in vertebrate galectins for phase separation" analyses how the IDR of hGal3 has evolved through vertebrate lineages. The authors find that all the vertebrate lineages show repeating motifs with similar sequence patterning that always includes an aromatic residue, and while the residue identity is conserved within the repeats, it is not between different lineages, indicating that the aromatic residue identity diverged prior to the repeating motifs expanding. They find that the aromatic residue is required for function, however the identity of the residue is not critical. They also find a region at the N-terminus of the IDR that is conserved across vertebrates that contributes to self-association by interacting with the folded CRD.

The work is interesting and should appeal to an increasingly broad audience of people interested in IDRs and how they mediate phase separation.

We thank the reviewer for the positive assessment of our work.

Major comments/questions:

The authors show that you can get similar behavior between the Zebrafish and human orthologs even though the Zebrafish has Tryptophan residues in its repeats while the human has Tyrosine. Oddly they find that substituting Tyrosine in place of Trp on the Zebrafish resulted in increased interactions between the CRD and NTD. This is interesting as Tryptophans usually have a stronger interaction strength than Tyrosine, as seen in their own zfGalau variant as well as in the literature (see 10.1091/mbc.E24-03-0128 and 10.1038/s41567-024-02558-1). Do the authors have any explanation for this?

We thank the reviewer for this comment. Indeed, tryptophan residues generally form stronger interactions than tyrosines, as also shown in many model peptide studies (cited in the manuscript: Wang 2018 and Bremer 2022). However, in our system, the interaction network involves not only aromatic contacts within the IDR but also electrostatic and cation- π interactions between the IDR and the structured CRD. Our previous NMR data (Sun 2024, PMID: 39248370) indicate that positively charged residues on the CRD's non-ligand-binding surface (the NTD-interacting site) also contribute significantly to self-association. Because the zebrafish CRD differs from the human CRD in both the number and positioning of these charged residues, substituting tyrosine for tryptophan in the zebrafish IDR likely alters the local cation- π balance rather than simply strengthening π - π interactions. This could explain why the tyrosine-substituted variant exhibits enhanced NTD-CRD interaction, despite the intrinsically weaker aromatic character of tyrosine. Furthermore, the patterning

of aromatic residues may also affect the level of assembly (cited in the manuscript, Martin 2020), and such patterning was not considered in our direct substitution of aromatic types. Nevertheless, our results suggest that the evolutionary optimization of galectin assembly arises from modulating motif repetition rather than selecting a single “strongest” aromatic residue.

We have included additional discussion in the third paragraph of the Discussion section to address this point. “One of the possible reasons may stem from the interactions between NTD/CRD. As demonstrated in our previous NMR analyses, positively charged residues on the CRD’s non-ligand-binding surface (the NTD-interacting face) contribute to overall assembly via cation- π interactions. Although the zebrafish CRD is conserved relative to the human ortholog, the number and spatial arrangement of these basic residues differ. Consequently, replacing tryptophans with tyrosines does not necessarily enhance assembly strength, as might be expected from model peptide studies. Instead, our data suggest that the evolutionary optimization of galectin assembly is achieved not by selecting the strongest individual “sticker” residue, but by duplicating the aromatic-residue-containing motif multiple times, which seems to be the more efficient evolutionary strategy.”

For Figure 6 (primarily, but this affects the other chemical shift plots as well) they show a line at 0.01PPM and treat shifts greater than this as significant, but they do not directly state that this is the digital resolution of the experiments. Furthermore, they do not give the number of increments for the experiments in the methods, so it is impossible of the reader to tell if this is significant or not. Additionally, the difference in relaxation rates between the 40 and 400uM measurements is quite close to the noise in the experiment (See 6A, and 6C). The case for this region being important would be made stronger if there was complimentary evidence. Perhaps NMR in Trans, (i.e. 40uM of CRD plus 400uM of the Nterm residues?). Or the solubility experiments in 4G, and 5F with a construct lacking this Nterm domain. Perhaps even adding the AlphaFold3 prediction showin an interaction between this N-terminal region and the CRD as well.

We thank the reviewer for these comments. The horizontal line at 0.01 ppm in Fig. 6 (and in the other chemical-shift-perturbation plots) is not the digital point spacing of the spectra. It is an arbitrary threshold we used to highlight large perturbations in the weighted HN chemical-shift perturbation (CSP), defined as (in the Supporting Text, Appendix):

$$\Delta\delta^{\text{HN}} = \sqrt{\frac{(\Delta\delta_{\text{H}})^2 + \left(\frac{1}{5}\Delta\delta_{\text{N}}\right)^2}{2}}$$

Under this metric, the enlarged K178 in Fig. 3A ($\Delta\delta_{\text{H}} = 0.018$ ppm, $\Delta\delta_{\text{N}} = 0.015$ ppm) gives $\Delta\delta_{\text{HN}} = 0.013$ ppm, for example; for Q228 ($\Delta\delta_{\text{H}} = 0.012$ ppm, $\Delta\delta_{\text{N}} = 0.136$ ppm) $\Delta\delta_{\text{HN}} = 0.021$ ppm. These cases show clear CSPs on the overlaid spectra despite modest weighted values, whereas many other residues have near-zero $\Delta\delta_{\text{HN}}$, indicating no CSP.

As suggested, we have added the acquisition/processing details to the

Supporting Text in the Appendix: “.....The ^{15}N dimension was sampled with 128 increments and zero-filled to 1024 points in NMRPipe prior to Fourier transformation; the ^1H dimension was acquired with 2048 points.....” and added “.....chemical shift perturbations greater than 0.01 ppm (an arbitrary threshold for indicating the most pronounced perturbations) are highlighted.....” into the figure legend to avoid confusion.

The concentration-dependent changes in R2 between 40 and 400 μM in panels 6A and 6C appear modest because we deliberately used a common y-axis across panels to avoid overstating small effects. Nevertheless, a consistent, reproducible per-residue increase is evident that is absent in the CRD-only control (Fig. 6B). To aid visualization while preserving a shared scale, the per-residue differences are shown as brown bars. Following Reviewer 3’s suggestion, we also quantify the construct-specific effect by $\Delta\Delta\text{R2} = \Delta\text{R2}(\Delta\text{22–113}) - \Delta\text{R2}(\text{CRD})$. This analysis, presented in the revised Figure 6D, demonstrates a significant increase in R2 for $\Delta\text{22–113}$ at higher concentration, consistent with weak self-association (statistical details are provided in the Methods/figure legend). For the other cases (Figs. 3G, 3H, and 4F), we compared CRD-averaged values across constructs; the results are summarized in the revised Table 1 and show significant differences.

Regarding the complementary evidence that the reviewer suggested, we examined constructs of $\Delta\text{22–113}$ for the solubility/precipitation test. Their precipitation behavior is very similar to the CRD-only construct, indicating that

the N-terminus (residues 1-21) is not the dominant driver of bulk solubility under our conditions. We have added the additional results in Appendix Figure S5 and mentioned in the revised Results: “...Although the agglutination ability of hGal3^{Δ22–113} was similar to the CRD-only construct (*Appendix*, Fig. S5), the R_{2S} of hGal3^{Δ22–113} still increased between protein concentrations of 40 and 400 μ M (Fig. 6A)....”

Appendix Figure S5. LPS-induced agglutination depends on the conserved galectin-3 N-terminus. Wild-type human galectin-3 (hGal3) is compared with two N-terminal variants: hGal3^{Δ22–113}, in which N-terminal domain (NTD) residues 22–113 are deleted, leaving only residues 1–21 N-terminal to the CRD, and hGal3^{Δ20}, in which NTD residues 1–20 are deleted, as well as the isolated carbohydrate-recognition domain (hCRD). (A) Coomassie-stained SDS–PAGE of the total reaction (T) and the corresponding supernatant (S) and pellet (P) fractions after incubation with LPS and centrifugation (conditions in Methods). (B) Protein in S and T quantified by Bradford assay (n = 3; mean \pm s.d.).

The NMR in trans approach was reported previously for this protein (Ippel et al., *Glycobiology*, 2016) but required a \sim 1:50 titration (\sim 1 mM N-terminal construct) to elicit detectable responses, consistent with a very weak N-terminal/CRD interaction. At the concentrations used here (40–400 μ M), trans experiments would be below sensitivity.

AlphaFold predicted human galectin-3 place segments of the N-terminus against the CRD surface (see the inserted figure). This prediction does not fit our previous works (2017 Lin JBC). In that work, we used N-terminal IDR systematically

truncated constructs to collect concentration-dependent CSP, dynamic analysis, and intermolecular PREs. Collectively, these results confirm widespread, low-affinity contacts between the N-terminal region and the CRD, consistent with a fuzzy interaction between the NTD and CRD. To avoid misunderstanding and in reply to the reviewer's first minor comment, we have summarized our previous works on galectin-3 in Figure EV1 and described them in the Introduction (see below reply).

But if this is true, how does this affect the model? If there is a specific interaction between this N-terminal region and the CRD, could it not act as an autoinhibitory element, a common feature in tethered IDRs? Given that they are tethered to one another the effective concentration would be quite high (likely in the mM range) perhaps increasing the concentration required for higher order assemblies/phase separation to occur. This could perhaps also explain the increased intramolecular interactions found in the zfGalW/Y variant. For literature on the effect of linker length on self/autoinhibitory-interactions there are extensive works by Magnus Kjaergaard and Lucia Chemes.

We appreciate the reviewer's thoughtful comment, which helps us realize that our original description of the galectin-3 self-association model may have been unclear. We have clarified this in the revised Introduction and in Figure EV1. Briefly, the NTD and CRD do not form a specific, high-affinity intramolecular complex; rather, they engage in fuzzy, dynamic contacts (Lin et al., JBC 2017). Importantly, the NTD interacts with the non-carbohydrate-binding face of the CRD, leaving the sugar-binding site accessible; therefore, the interaction does not constitute autoinhibition. Although tethering can increase the effective intramolecular concentration, the absence of a defined interface and the preservation of the ligand-binding site argue against an autoinhibitory mechanism in our system. The increased intramolecular NTD-CRD interaction observed in the zfGalW/Y variant is more likely explained by differences in the positively charged side-chain composition on the CRD and/or the distribution of aromatic residues within the NTD, as detailed in the above reply, which we have also addressed in the Discussion. We hope these clarifications resolve the concern.

Minor Comments/Questions:

There is still an ongoing thread in the literature suggesting that hGal3 functions as a dimer. The authors addressed/refuted this in their earlier work, but perhaps it is still necessary to repeat that.

We thank the reviewer for this suggestion. To prevent confusion in interpreting our results, we now include a brief background paragraph that summarizes our prior findings on hGal3's oligomeric (not a dimer) behavior. We also added a schematic overview (Figure EV1) that summarizes the key findings from these studies. We hope this clarification will help general readers to follow our reasoning and results.

“.....In our previous biophysical and structural studies, we showed that NTDs self-associate and make fuzzy contacts with the CRD's non-carbohydrate-binding site (hereafter, the NTD-binding face) in both intra- and inter-molecular modes (Figure EV1). These interactions drive galectin-3 phase separation via π - π interactions and explain that the protein's agglutination arises not from dimeric or tandem CRD repeats but from multivalent NTD-mediated association (Figure EV1). We further assessed cation- π interactions contributed by conserved, positively charged residues at the NTD-binding face (Figure EV1). This detailed mechanistic framework makes galectin-3 a suitable model for our analysis. In this work,

Figure EV 1. Summary of previous studies on galectin-3 self-association. Background shading groups panels by publication: yellow: JBC (2017); peach: Nat. Commun. (2020); blue: Adv. Sci. (2024). Galectin-3's intrinsically disordered N-terminal domain (NTD) forms transient intra- and intermolecular contacts with the non-carbohydrate-binding face of its carbohydrate-recognition domain (CRD; the NTD-binding face) and also self-associates (yellow). These interactions drive phase separation (PS) and enable galectin-3-mediated

agglutination, assessed with lipopolysaccharide (LPS) micelles. Agglutination is supported by π - π interactions among aromatic residues within the NTD (peach) and by cation- π interactions between these aromatics and conserved, positively charged residues on the NTD-binding face. Two conserved acidic residues in the NTD modulate self-association in a pH-dependent manner (blue).

Figure 1.

1E, can the authors provide a legend of what the different colored modules in the IDR denote.

In this figure, the repeated colors are used only to indicate that the motifs are identical within the same species but differ among orthologs. The detailed motif sequences are provided in Dataset EV4, which is presented as an interactive HTML file. One can also access through the following link:

https://jierongh.lab.nycu.edu.tw/ED_4_tetrapoda_fish_tp_meme.html

(Access to this link is anonymous and not tracked.)

Hovering the cursor over each motif in this file displays its sequence. Therefore, we believe this supporting data is clear, and adding legends directly to Figure 1 would not be informative.

Figure 2/7.

2F, 7ii. Model of galectin interactions leading to PS, it looks like the folded domain/NTD, is main only interaction, shouldn't there also be some, or perhaps better indication that the aromatics within the IDR are adding to the interactions.

We appreciate the reviewer's observation. These schematic figures are intended to illustrate the overall interaction framework, but the small scale may have made it unclear that both NTD-CRD and NTD-NTD interactions contribute to agglutination. We did include some indication of IDR contacts, although they may not be visually apparent. In the revised Figure EV1 and Introduction, we now clarify that aromatic residues within the IDRs also participate in these multivalent interactions. We believe the updated description and figure resolve this ambiguity.

Referee #2:

The manuscript entitled "Evolution of intrinsically disordered regions in vertebrate galectins for

phase separation" by Chen et al. presents a deep bioinformatics and structural investigation of several Galectin-3 forms to decipher the role of the composition of the IDR in intramolecular interactions, self-association and phase separation. The authors show that the IDR is responsible of these functions regardless of the specific aromatic amino acids present in these tails. Using NMR they unambiguously show that tryptophan and tyrosine, although with different intensity, play a fundamental role in these functions. Indeed, Modulation of the self-interaction and agglutination are mainly dictated by the number of aromatic-containing repeats more than the amino acid. Finally, the authors identify a highly conserved segment at the N-terminus of the IDR that plays a fundamental role in self-association.

The study is very elegant and properly done and results are clearly described. Conclusions are solid and in accordance with the results obtained. In my opinion, the authors convincingly disentangle the functional role of aromatics in galectin-3, and present a model that can be probably applied to many other protein families.

We appreciate that the reviewer thought highly of our work.

In my opinion, the functional role of the conserved N-terminal segment in self-association and intra-molecular interactions would be reinforced if a mutant without this fragment would be studied. This would reinforce the image (maybe simplistic) that this region is the only responsible of the intramolecular interactions.

We thank the reviewer for this suggestion. We have created a construct without the first amino acids (denoted as hGal3^{Δ20}). In the Appendix Figure S5 (also see the reply to Reviewer 1's major comment), the agglutination level of this construct is similar to that of the full-length (hGal3). These results fit with our model because this conserved fragment in all vertebrates only provides marginal self-association capability (Figure 6). It is our proposed model in this article that this marginal self-association ability is the potential reason why this galectin does not evolve as a dimer or tandem repeat but instead uses aromatic-residue-containing motifs to increase its self-association ability.

We have also recorded NMR data at high and low concentrations (400 μ M and 40 μ M) of hGal3^{Δ20}. The dynamics and chemical shift perturbation between these two concentrations are similar to those of the wild-type. We do not think including or mentioning these NMR results would enhance the clarity of our conclusion.

Referee #3:

Comments for EMBOR-2025-62185V1

Title: Evolution of intrinsically disordered regions in vertebrate galectins for phase separation

This manuscript provides a proteome-wide analysis of the evolution of intrinsically disordered regions (IDRs) in the galectin-3 family, investigating their roles in driving phase separation (PS) and agglutination. Building on their 2020 Nature Communications study-where they demonstrated that human galectin-3 (hGal3) uses PS to agglutinate LPS micelles-the authors now extend their investigation to zebrafish galectin (zfGal). They integrate sequence mining of over 1,300 galectin-3 homologs, computational predictions (disorder propensity, PS scoring, motif enrichment, π - π interaction potential), and targeted biophysical experiments (NMR HSQC and R_1/R_2 relaxation, circular dichroism, and LPS-based turbidity assays). By systematically comparing zfGal and hGal3-and swapping aromatic motifs and numbers-they dissect the relative contributions of motif number versus aromatic residue identity to agglutination, concluding greater importance for the **former** feature.

Overall, the manuscript is interesting and worthy of publication in EMBO Reports. There are several experiments and revisions the authors could do to improve the manuscript, which I list below:

We thank the reviewer for thinking our work interesting and worthy of publication.

Major Comments (roughly in order of importance)

1. Confusing Logic

I found the logic of the experiments and the choices of comparisons hard to follow, making the authors' conclusions difficult to understand. Often apples are compared to oranges (e.g. R2 of hGal3 to zfGal3aug, which are hard to relate due to differences in aromatic residue type, IDR patterning and the CRD sequences), and apples to apples comparisons that should have been shown are not (e.g. R2 of zfGal3 WT to W/Y to aug, which show clear trends of self-association that parallel agglutination trends but are shown in different figures; also R2 of hGal3 vs hGal3WY/G, which also show patterns that parallel agglutination but are also shown in different figures). I would suggest a rearrangement of some of the data to make the relationships between sequence, R2 and agglutination easier to understand. Relatedly, **a summary table of average R2 values for the CRD residues** of the different proteins would facilitate comparison.

We thank the reviewer for highlighting the confusion regarding our coloring choices and figure logic. We have carefully considered these points and made adjustments to improve clarity.

We agree that consistent and intuitive coloring is important for readability. In our figures, we use a “warmer” palette (e.g., orange and related shades) for the fish variants, except for zfGal^{W/Y}, which is rendered in light green to reflect its humanized tyrosine motif. Human variants are shown in blue or blue-purple. To ensure consistency throughout the manuscript, we have revised the color of Figure 6 to purple, matching the HSQC in Figure 3B.

For NMR dynamics analysis, we use red to indicate high concentration (400 μ M), black for low concentration (40 μ M), and brown bars for the differences between these conditions. To avoid confusion, we have clarified all figure legends, explicitly indicating the color code for each construct or experimental condition (see the revised Figures 3A, 3B, 3G, 3H, 4B, 4C, 4F, 6A–C).

Our figure arrangement follows a logical progression: (1) We first show the effect of the IDR on dynamics in zfGal (Fig. 3G vs 3H). (2) We then demonstrate that the W to Y mutation has a limited effect on the dynamics (Fig. 4F; a common y-axis across dynamics data for comparison, e.g., to 3G). (3) Finally, we show that the number of repeated motifs has a greater impact (Fig. 5).

We believe this order best describes our findings, and rearranging the figures would not make the logic clearer. However, as the reviewer suggested, we have added a summary table of average dynamics values for direct comparison between different constructs and variants. The new table in the manuscript:

Table 1.

Construct	zfGal		zfCRD		zfGal ^{W/Y}	
Conc. (μ M)	40	400	40	400	40	400
R_2	17.03 \pm 0.29	21.05 \pm 0.31	10.05 \pm 0.14	10.97 \pm 0.17	18.83 \pm 0.36	22.73 \pm 0.49
$ \Delta R_2 $	4.03 \pm 0.43		0.92 \pm 0.22		3.90 \pm 0.61	
R_1	0.82 \pm 0.01	0.72 \pm 0.01	1.48 \pm 0.01	1.41 \pm 0.01	0.76 \pm 0.01	0.65 \pm 0.01
$ \Delta R_1 $	0.10 \pm 0.01		0.07 \pm 0.01		0.10 \pm 0.01	
R_2/R_1	20.65 \pm 0.46	29.26 \pm 0.54	6.79 \pm 0.10	7.75 \pm 0.13	24.90 \pm 0.53	34.87 \pm 0.84
$ \Delta (R_2/R_1) $	8.61 \pm 0.71		0.96 \pm 0.16		9.97 \pm 1.00	

Dynamic analysis for the indicated constructs corresponding to Fig. 3G (zfGal), Fig. 3F (zfCRD), and Fig. 4F

(zfGalW/Y). Only residues in the carbohydrate-recognition domain (CRD) were analyzed. For each condition (40 μ M, 400 μ M), R_2 and R_1 are inverse-variance weighted (IVW) means of the residue-wise estimates; standard errors (SEs) are the IVW SEs. ΔR_2 and ΔR_1 are the direct differences of the condition means. R_2/R_1 is the ratio of the IVW means at the same concentration, and $\Delta(R_2/R_1)$ is the difference between concentrations. Uncertainties for R_2/R_1 , ΔR_2 , ΔR_1 , and $\Delta(R_2/R_1)$ are obtained by standard error propagation from the reported SEs of the IVW means.

We appreciate the reviewer for these suggestions, which have helped us improve the clarity and presentation of our results.

2. Missing Reciprocal Mutants

The authors swap W \rightarrow Y in zfGal to mimic hGal3, but do not analyze the reverse (Y \rightarrow W) in hGal3 and compare it to zfGalaug. Nor do they make a zfGalaug-W/Y mutant to compare with zfGalaug. Testing these mutants would better confirm whether residue identity truly has minimal impact and are key symmetry checks.

We thank you for the suggestions regarding reciprocal mutants. As we noted in our reply to Reviewer 1, it is not our intention to overturn or re-examine the effects of Y versus W, particularly regarding the cation- π interactions contributed from human or fish CRD in addition to the π - π from the NTDs, which make the agglutination mechanism of these IDR-tethered galectins complicated and are beyond the scope of our current study.

Our central argument is that natural selection does not always favor the strongest single interaction; rather, self-association strength can be tuned by varying the number of aromatic residues. This is exemplified in zfGal^{aug}, where the increased repeat number leads to increased self-association compared to zfGal. We do not think constructing a zfGal^{aug} W \rightarrow Y mutant is necessary, as the outcome (whether increased or reduced association) would not affect our main conclusions regarding the evolutionary strategy (increased motif number vs. residue identity).

However, as the reviewer suggested, we constructed and purified the hGal3^{Y/W} construct (replacing ten Ys with Ws in hGal3). Notably, at a high concentration (400 μ M, 303 K), this variant showed much more pronounced self-association, as evidenced by solution turbidity (the wild-type sample is transparent in the identical conditions) and significantly reduced HSQC cross-peak intensities (see *Panel A* in the figure below). Interestingly, chemical shift perturbations indicating the NTD/CRD interactions (e.g., at residue 216) are nearly abolished, differing from the wild-type (see *Panel A* in the figure below; also see Figure 3B in the manuscript and referring to our previous work: Lin et al. JBC 2017). On

the other hand, at lower concentration (40 μ M), the NTD/CRD interaction persists, more similar to the wild-type construct (*Panel B* in the figure below). Thus, W substitutions in the NTD strengthen NTD/NTD interactions to an extent that overrides NTD/CRD interactions (as illustrated in *Panel C* in the figure below). In contrast, at a lower concentration, the hGal^{Y/W} construct's NTD/CRD interaction would be more similar to the wild-type (*Panel D* in the figure below).

These findings support our argument, also in the response addressed to Reviewer 1: NTD/CRD interactions play a critical role in modulating self-association, making it inappropriate to focus solely on aromatic side-chain strength in these IDR-tethered galectin systems. The interplay between NTD/CRD is an intriguing subject, but it lies beyond the scope of our present manuscript. The theme of this manuscript is that, regardless of aromatic type strength, evolutionary selection may favor motif duplication rather than simply optimizing sticker strength.

As we explained above, we also think discussion of the hGal3^{Y/W} construct would distract from the central logic of the manuscript; we include these data here only for the reviewer's reference, shown in the figure below. We hope this clarifies our rationale and approach.

3. Intra- vs intermolecular interactions

It is unclear how the authors know that the chemical shift perturbations in the various figures result from intramolecular interactions. No KD value is stated for the interactions, and even at 40 μM there could be intermolecular interactions that perturb chemical shift. The authors should examine interactions between the IDR and CRD in trans to assess this possibility, and/or measure the relevant KD values.

In our previous study (Lin et al. JBC 2017), we systematically investigated the self-association behavior of galectin-3 using a combination of concentration-dependent NMR dynamics, chemical shift perturbation (CSP), and intermolecular paramagnetic relaxation enhancement (PRE) experiments with various NTD-truncated constructs. Specifically, by studying high and low protein concentrations and using constructs both with and without the IDR, we were able to map the sites involved in intra- and intermolecular interactions (see also *Panel D* in the figure in our previous response).

At 40 μM , our intermolecular PRE experiments (using MTSL-labeled ^{14}N -galectin-3 mixed 1:1 with ^{15}N -galectin-3) showed negligible line broadening, indicating that intermolecular interactions are minimal under these conditions (see Figure 5 in Lin et al. JBC 2017). Thus, the CSPs we observe at 40 μM can be attributed predominantly to intramolecular NTD/CRD contacts (when comparing full-length and CRD-only).

Consistent with this, the KD for the NTD/CRD trans interaction is in the millimolar range as demonstrated by Ippel et al. (Glycobiology 2016). The intermolecular interactions are extremely weak at the concentrations used for our main analyses. Therefore, knowing the precise KD does not affect our conclusions or the assignment of CSPs to intramolecular effects.

For clarity, we have added the following sentences immediately before the concentration-dependent NMR experiments. “To probe self-association, we compared concentration-dependent NMR spectra and dynamics to shift the monomer-oligomer equilibrium. Our earlier work showed that intermolecular interactions are negligible at 40 μM , whereas self-association is markedly increased at 400 μM without entering the PS regime. Consistent with this...” Also, the relevant background is provided in the Introduction and Figure EV1 in the revised manuscript (see also our reply to Reviewer 1). We hope this addresses the concerns regarding the distinction between intra- and intermolecular effects in our system.

4. Interactions of the N-terminal Motif

The conclusion from figure 6 is confusing compared to earlier data. In figure 6, chemical shift perturbations of the CRD are attributed to interactions with the N-terminal 1-21 motif. This motif does not have either W or Y residues, but earlier data show that mutating W to Y in zfGal3 changes CRD chemical shift perturbations due to the IDR. How to reconcile the two observations?

We apologize if our interpretation was unclear. The goal of Figure 6 is to address a separate but complementary question from the earlier figures: Why is a short, conserved motif present at the N-terminus of all vertebrate IDR-tethered galectins (as shown in Fig. 1D and 1G), even though it lacks aromatic residues like W or Y?

Earlier in Figures 4 and 5, we showed that: (1) The identity of aromatic residues (W vs. Y) has only a modest effect on self-association. (2) The number of repeat motifs has a more pronounced effect, especially in constructs like zfGal^{aug}.

Then, in the following Figure 6, we shifted our focus to the evolutionarily conserved N-terminal residues 1–21 of hGal3, which do not contain aromatic residues. Despite this, our data show that this fragment still contributes measurably to self-association, likely electrostatic or polar contacts, especially in early stages of IDR evolution of this type of galectin.

We discuss this idea in the 5th and 6th paragraphs of the Discussion. To clarify this point in the manuscript, we have also revised the Results section describing Fig. 6 with the following text to improve the logic flow.

“Apart from the repeated motifs on the same species, the motif analysis also revealed that the N-terminal residues of the IDRs are conserved across vertebrates (Fig. 1G). To investigate the role of this conserved fragment, we constructed a human CRD tethered to the first 21 residues of the NTD.....”

We hope these clarifications resolve the confusion.

5. Lack of Direct Condensate Imaging

The authors do not show DIC/fluorescence (or FRAP) images of zfGal or zfGal^{W/Y} (+ LPS) droplets-only turbidity and NMR. They should include images (and FRAP kinetics if possible), ideally alongside their own hGal3 images from 2020, to unambiguously demonstrate liquid like phase separation of the zebrafish proteins.

The zfGal constructs (zfGal and zfGal^{W/Y}) exhibit significantly reduced self-association than human galectin-3 (as shown in NMR analysis), and thus, they

do not condense solely as human galectin-3 under the conditions we have used. In our previous work (2020 Nature Communications), we performed FRAP on human galectin-3 condensates at high protein concentrations (above 500 μ M) in the presence of high salt (above 500 mM) to demonstrate their liquid-like behavior. However, these conditions are far from physiological and were intended only to highlight the dynamic potential of the system. We then instead used the LPS micelle model, a well-established system in the galectin field, to evaluate agglutination ability and to verify its reversibility by adding an excess amount of its ligand (lactose) to disrupt the interaction (depicted in Figure 2F).

This reversible transition is a key feature of galectin's functional PS and serves to demonstrate dynamic behavior. Given this, we think that FRAP or additional imaging under these conditions would not provide further mechanistic insight and is not necessary for the conclusions drawn about zfGal or zfGal^{W/Y}.

6. Quantitative π - π Scores & Aromatic Composition

The authors quote PScore/FuzDrop but do not summarize those interaction scores or % of F/Y/W across their 1,321 hits or filtered 351 "hGal3 like" set. A heatmap or boxplot of π - π scores and aromatic percentages would help evaluate their evolutionary claims.

We appreciate the reviewer's suggestion to provide quantitative statistics for aromatic residues and π - π interaction scores across our dataset.

As noted, we used PS score and FuzDrop specifically to illustrate phase separation tendencies in the representative human and zebrafish sequences (Fig. 2E), as these scores closely reflect the level of structural disorder (Fig. 2A) and prion-like propensity. We provided the full set of IUPRED results for all 1,321 hits in **Dataset EV2**, and chose not to plot PS score or FuzDrop distributions because their overall patterns are similar to the level of structural disorder, and these predictors are local, sequence-dependent, and not easily summarized as a single meaningful value across diverse sequences.

In response to the reviewer's main point, we have now quantitatively analyzed the distributions of W, F, and Y residues within the IDRs (their populations are shown in **Dataset EV3** as pie charts). We have summarized these statistics in a new Fig. 3C, which presents boxplots of aromatic residue frequencies within the IDRs across the four vertebrate groups, alongside representative pie charts for each taxon. These data reinforce our evolutionary

conclusions: fishes display a broader distribution of all three aromatics (W, F, Y) within their IDRs, while mammals are notably enriched in Y. The corresponding figure legend has been updated as follows: “(C) Boxplots show the distributions of W, F, and Y content (percent of IDR) in galectin homologs across major vertebrate clades (median and interquartile range shown; whiskers denote data spread). Representative pie charts further illustrate the aromatic composition per group (W: tryptophan, F: phenylalanine, Y: tyrosine; all data in Dataset EV3).” And in the main text: “Fig. 1C shows the distributions of %W, %F, and %Y in IDRs across vertebrates, alongside representative pie charts from Dataset EV3”

We hope this new analysis provides a clear quantitative foundation for the evolutionary patterns discussed in the manuscript, as suggested by the reviewer.

Minor Comments

1. No Color Legends in Key Figures

Panels like Fig 3A/B, 3G/H, 4E/F lack legends. The authors should consider putting label legends "black = 40 μ M; red = 400 μ M" and "green = CRD-only; blue = full-length" in the figures in addition to captions for visual ease of the reader.

We thank the reviewer for this suggestion. We have added explicit color legends directly within the relevant figure panels, including Figures 3A, 3B, 3G, 3H, 4B, 4C, 4F, and 6A–C. We hope these additions make the data presentation more accessible and self-contained for the readers.

2. No Direct hGal3 vs. zfGal^{W/Y} Visuals in Fig 4

Per major point 1 above, the authors present zfGal^{W/Y} data but never juxtapose it with hGal3. A side by side panel for HSQC overlays and LPS assays would highlight the modest effect of aromatic type.

As discussed in our response to major point 2 above, a direct visual comparison between hGal3 and zfGal^{W/Y} (or hGal3^{Y/W} and zfGal) would not be meaningful, because the human and zebrafish CRDs differ substantially and introduce different self-association modes. The Y/W substitutions are evaluated within each species to examine their effects, rather than for cross-species equivalence. In our *Appendix* Figure S2, we have compared the full-length and CRD-only HSQC spectra of zfGal and hGal3. They are very different, as expected. Therefore, in our opinion, comparing hGal3 and zfGal^{W/Y} HSQC would not

provide additional mechanistic insight and might be misleading.

3. Undefined "Weak" vs. "Strong" Effects

The authors call aromatic-type effects "weak," but never set thresholds. They should define, e.g., $\Delta\delta < 0.05$ ppm as weak or specify a % change in R_2 that counts as significant (relative to errors in R_2).

We agree that defining specific thresholds can be useful in some contexts. However, in our study, we interpret chemical shift perturbations ($\Delta\delta_{\text{HN}}$) and relaxation changes (R_2 , R_1 , R_2/R_1) in a relative manner, comparing across residues and experimental conditions within the same system. This approach allows us to identify regions with significant perturbations or changes without arbitrarily assigning cutoffs. For example, in Fig. 4D and 4E, regions with the largest $\Delta\delta_{\text{HN}}$ values stand out clearly, and similarly, in Fig. 4G and 4H, regions of notable R_2/R_1 elevation are clear and discussed accordingly. We used the terms "weak" and "strong" descriptively to reflect relative trends, supported by the global pattern of changes rather than an absolute threshold.

However, we agree that the terms "strong" and "weak" are misleading, and we have revised wordings throughout the manuscript to avoid ambiguous labels and emphasize relative magnitude. For example, "...indicating a **greater** tendency toward..."; "... the **increased** agglutination ability..."; "...these constructs are **most pronounced** in..."; "...chemical shift perturbations are **less pronounced** in..."; "...has **reduced** agglutination capacities..."

4. No Statistics for R_2 Increase in $\Delta 22-113$

The $\Delta 22-113$ construct shows a small R_2 bump vs. CRD alone but lacks error bars or significance tests. A simple mean {plus minus} SD or t test would clarify if this is a real effect.

We thank the reviewer for this suggestion. We have performed a statistical analysis of the per-residue difference-of-difference in relaxation rates ($\Delta\Delta R_2$) between the $\Delta 22-113$ construct and the CRD alone, as shown in the **new Figure 6D**. [$\Delta\Delta R_2 = \Delta R_2(\text{hGal3}^{\Delta 22-113}) - \Delta R_2(\text{hCRD})$]

Rather than relying solely on mean \pm SD or a t-test, we analyzed all paired $\Delta\Delta R_2$ values across residues. Specifically, the majority (81%) of $\Delta\Delta R_2$ values are positive, and a one-sided sign test rejects the null hypothesis that the median $\Delta\Delta R_2$ is zero ($p = 7.1 \times 10^{-12}$). The median increase is 0.74 s^{-1} , with a 95% confidence interval $[0.55, 0.95] \text{ s}^{-1}$ estimated by bootstrap resampling. Accordingly, the observed R_2 enhancement in $\Delta 22-113$ construct is a true

effect and not attributable to random noise or variability.

We have added “...Per-residue difference-of-differences in transverse relaxation rate [$\Delta\Delta R_2 = \Delta R_2(\text{hGal3}^{\Delta 22-113}) - \Delta R_2(\text{hCRD})$] are mostly positive and a one-sided sign test rejects the null hypothesis that the median $\Delta\Delta R_2$ is zero (Fig. 6D). These analyses indicate that the observed R_2 enhancement in $\text{hGal3}^{\Delta 22-113}$ construct is a true effect and not attributable to random noise or variability, suggesting...” in the Results. Their statistical significance is now presented in the revised Figure 6D and detailed in the figure legend.

“...(D) Per-residue difference-of-differences in transverse relaxation rate [$\Delta\Delta R_2 = \Delta R_2(\text{hGal3}^{\Delta 22-113}) - \Delta R_2(\text{hCRD})$]. The histogram (right panel) shows the distribution of $\Delta\Delta R_2$ values. A one-sided sign test rejects the null hypothesis of no shift ($p = 7.1 \times 10^{-12}$)...”

The script for the analysis is described and deposited in Source Data.

5. Overstated "Motif Duplication" Mechanism

The authors' claims that they "demonstrate" evolution via motif duplication is too strong based on a few example species. They need broader phylogenetic statistics on motif copy number changes or should soften wording to "consistent with" duplication as one plausible route.

We have now toned down the last sentence in the first paragraph in the Discussion to “...While there may be many pathways for IDRs to evolve, motif duplication would represent a relatively straightforward route, *consistent with* our observation for galectins.”

6. Delineation of interaction types

The authors should do more in their analysis to delineate the roles of IDR-IDR and IDR-CRD interactions in promoting agglutination. E.g. in the zfGal^{W/Y} mutant, IDR-CRD interactions increase, but agglutination decreases, in apparent contradiction (Fig. 4). This contradiction is resolvable, though, if one assumes that the IDR-IDR interactions would be weakened by the W→Y mutations, as seen with other phase separating systems. It is possible that the overall weak dependence of agglutination on aromatic residue type reflects trade offs between IDR-IDR and IDR-CRD interactions. This is worth discussing. In general, the authors should do more to distinguish the different types of interactions reported on in their data. Intermolecular vs intramolecular, IDR-IDR vs IDR-CRD, self-assembly vs agglutination.

We thank the reviewer for this comment. As suggested by the reviewer to investigate the hGal3^{Y^W} construct (discussed in major comment 2) and our zfGal^{W/Y} mutant, the W or Y substitutions do alter the population of NTD/CRD versus NTD/NTD interactions. The different spacing of aromatic residues in fish and human NTDs also influences the regulation of these interactions. In addition, the charged side-chain composition and spatial distribution on the CRD's NTD-binding face, which differ between zebrafish and human, also affect these interactions. To address these issues, tethering zfNTD to hCRD and vice versa could be informative. Previously, we created approximately 10 mutants on the hCRD's NTD-binding face with varying charge levels to modulate NTD/CRD and NTD/NTD interactions (see Sun et al. 2024 Adv. Sci.). By combining all these mutants with NMR and small-angle X-ray scattering (reflecting the “openness” of NTD/CRD), we could quantitatively reveal different interaction modes. Disentangling the respective contributions of NTD/CRD and NTD/NTD interactions to self-association is indeed an ongoing effort in our lab, but it lies beyond the scope of this manuscript. We believe our current conclusions on the evolutionary route of galectin IDRs remain valid without such detailed dissection.

Dear Prof. Huang

Thank you for the submission of your revised manuscript to our offices. We have now received the enclosed reports from two of the original three referees that were asked to assess it. While they were both satisfied with your revision, EMBOR-2025-62185V2 still has minor issues that were flagged by editorial assistance team and must be fixed before I can formally accept your manuscript. Please look on the comments below and make sure to address them. Please note your figures will be re-checked for scientific integrity after resubmission (this is our normal procedure done for any manuscript before final acceptance).

I look forward to seeing a new revised version of your manuscript as soon as possible.

Yehu Moran
Academic Editor
EMBO Reports

Comments by editorial assistance team

MANUSCRIPT FORMAT: NOT OK - has figures; main and EV figures should be removed from the manuscript and each must be uploaded as a separate production quality Figure file.

COI/DCIS: in, but it needs to be renamed to Disclosure and Competing Interests Statement and should go after Acknowledgments.

AC/CRedit: needs to be removed from the manuscript text and appear only in our submission system.

CHECKLIST: included but missing corresponding author name, manuscript ID and journal. Please correct.

FUNDING INFO: not congruent; missing in the submission system: 110-2113-M-A49A-504-MY3; Yen Tjing Ling Medical Foundation (CI-110-16 and CI-111-19), the Higher Education Sprout Project by the Ministry of Education (MOE) in Taiwan; Taiwan International Graduate Program (TIGP) Rising Star Fellowship; the funders in the Comments box need to be removed and need to be provided via the separate entries.

FIGURES IN SEPARATE FILES: NO, please see above and correct.

FIGURE CALLOUTS: "Supporting Text" is not a correct callout. Please correct.

DATASET EV LEGENDS: Dataset EV1-EV4 uploaded as Related Manuscript Files; Dataset EV1-EV3 are in PDF which is not a correct format for a dataset; Dataset EV4 is a zip folder that has a html - better use a different format, if possible; the legends of the datasets are in the Appendix file which is not ok, each needs to be provided in its dataset file.

APPENDIX FILE WITH Table of Contents: in, but it is in Word format; we need clean PDF (no mark ups) and the table of contents needs to have each figure listed with its page number; there is also some text in the Appendix

SYNOPSIS IMAGE: missing, please provide.

SYNOPSIS TEXT: missing, please provide.

R&T TABLE: missing, please provide.

SOURCE DATA: Source Data provided with completed checklist; Source Data folders need to be uploaded separately.

EXTRA NOTES:

- Material and Methods should be renamed to Methods
- Table 1 and figure legends (main and EV) should be placed at the end, after References
- "This PDF file includes:" section should be removed from the manuscript.

** Figures are merged. Team will need to check figures when individual files are provided.

DATA CHECK: SUCCESS

*Please note that the specific URLs for dataset (Biological Magnetic Resonance Bank (BMRB): 52445.) is not provided in the data availability statement.

Figure Legends - Comments

- Please note that information related to n is missing in the legends of 1c; 3g,h; 6a-c. Please provide in figure legend.
- Please note that the error bars are not defined in the legends of figures 3g,h; 6a-c. Please define in the figure legend.

Comments by referees

Referee #1:

The authors of "Evolution of intrinsically disordered regions in vertebrate galectins for phase separation" have adequately addressed my previous concerns. I thank them for their efforts to explain in greater detail the intricacies of their system. I have learned something new in how to think about the interplay of dynamic complexes from our correspondence. I think the paper is

easier to understand following revisions, and I recommend the paper for acceptance.

Referee #2:

The authors have addequately addressed my points.

All minor editorial requests have been addressed by the authors.

Prof. Jie-rong Huang
National Yang Ming Chiao Tung University
Institute of Biochemistry and Molecular Biology
Taiwan

Dear Prof. Huang,

I am pleased to inform you that your manuscript has been accepted for publication in EMBO Reports. Your manuscript will be processed for publication by EMBO Press. It will be copy edited and you will receive page proofs prior to publication. Please note that you will be contacted by Springer Nature Author Services to complete licensing and payment information.

You may qualify for financial assistance for your publication charges - either via a Springer Nature fully open access agreement or an EMBO initiative. Check your eligibility: <https://link.springer.com/journal/44319/how-to-publish-with-us>

Yours sincerely,

Yehu Moran
Editor
EMBO Reports

>>> Please note that it is EMBO Reports policy for the transcript of the editorial process (containing referee reports and your response letter) to be published as an online supplement to each paper. If you do NOT want this, you will need to inform the Editorial Office via email immediately. More information is available here: <https://link.springer.com/partners/embo-press/editorial-policies#Peer%20review>